# The PIK3CA/AKT pathway drives therapy resistance in rhabdomyosarcoma

Qiqi Yang [1,2,3,4,6], Yueyang Wang[1,2,3,6], Luis A. Corchete Sanchez[2], Sabateeshan Mathavarajah[1,2,3], Qian Qin[1,2,3,5], Yun Wei[1,2,3], Eric Alpert[1,2,3], Lauren Whelton[1,2,3], Priyanshu Sharma [1,2,3], Stephanie Strom[1,2,3], Ilyas Oultache[1,2,3], A. John Iafrate[1,2], Luca Pinello [1,2,5], Esther Rheinbay[2], Chuan Yan[1,2,3,4] ✉ & David M. Langenau [1,2,3] ✉

Olaparib and temozolomide (OT) combination therapy is in clinical trial evaluation for rhabdomyosarcoma (RMS). Unfortunately, OT resistance has been reported in other cancers. Using preclinical mouse xenograft experiments, we show that OT effectively suppresses RMS growth, yet over half of RMS tumors develop resistance associated with transcriptomic changes that occur in the absence of recurrent genomic mutation. Importantly, most resistant RMS models upregulate the PIK3CA/AKT pathway, activating NRF2 phosphorylation and subsequent transcriptional expression of multidrug resistance ABC transporters. PIK3CA inhibitor alpelisib re-sensitizes resistant cells to OT by suppressing expression of ABC transporters. The combination of OT + alpelisib also kills RMS cells which are resistant to standard-of-care combination chemotherapy and was effective in preclinical xenograft mouse models at curbing tumor growth. Our work defines a common resistance pathway in RMS and has credentialled PIK3CA/AKT inhibition as a preclinical strategy to kill therapy resistant RMS.

Rhabdomyosarcoma (RMS) is a devastating pediatric cancer of muscle and is classified into three major subtypes. Fusion-positive RMS (FP-RMS) have DNA translocations that juxtapose PAX3 or PAX7 with the FOXO1 gene, creating a chimeric transcription factor that drives oncogenesis (PAX3:FOXO1 or PAX7:FOXO1). By contrast, fusion-negative RMS (FN-RMS) lack translocation driver events and are largely driven by RAS pathway activation[1–5]. Finally, spindle cell/sclerosing rhabdomyosarcoma (SS-RMS) are rare, can be highly aggressive and largely contain NCOA2::VGLL2 translocations or the MYOD^L122R mutation with associated RAS pathway activation[6,7]. For newly diagnosed RMS patients, front-line chemotherapy includes vincristine, actinomycin D, and cyclophosphamide (VAC), which is often combined with radiation and/or surgical resection[1]. Despite overall good responses to this intensive therapy regimen, 30% of patients eventually

relapse due to developing VAC resistance that is associated with an abysmal 17% five-year survival rate and occurs irrespective of disease subtype[8]. Therapy resistance has been linked to overexpression of ATP binding cassette (ABC) transporters that efflux drugs from the cell, leading to multidrug resistance (MDR) in other cancers; yet the mechanisms governing the possible heightened expression of these ABC drug efflux pumps in RMS are largely unknown[9,10]. Moreover, ABC transporter inhibitors have had little clinical efficacy, largely due to the functional redundancy and common upregulation of the family of 48 ABC transporters in resistant cancers. This makes it difficult to therapeutically target the wide array of ABC transporters that are co-expressed together and drive therapy resistance[11,12].

Combination therapies using the PARP inhibitor olaparib and the DNA-damaging agent temozolomide (OT) have demonstrated

[1]Molecular Pathology Unit, Massachusetts General Hospital, Charlestown, MA, USA. [2]Krantz Family Center for Cancer Research, Massachusetts General Hospital, Charlestown, MA, USA. [3]Harvard Stem Cell Institute, Cambridge, MA, USA. [4]Institute of Molecular and Cell Biology, Agency for Science, Technology, and Research (A*STAR), Singapore, Singapore. [5]Broad Institute of MIT and Harvard, Cambridge, MA, USA. [6]These authors contributed equally: Qiqi Yang, Yueyang Wang. ✉e-mail: yan_chuan@a-star.edu.sg; dlangenau@mgh.harvard.edu

exceptional responses in a variety of cancer types in both the preclinical and clinical trial settings. For example, OT treatment reduced tumor growth and prolonged survival of mice xenografted with Ewing's Sarcoma and RMS (Control ~40 days vs OT ~100 days)[13]. More importantly, OT treatment response for uterine leiomyosarcoma and small cell lung cancer patients was 27% and 41.7%, with median progression free survival of 6.9 and 4.2 months[13–18]. Indeed, our group has shown that OT was effective in curbing RMS tumor growth in zebrafish and mouse xenografts following three cycles of treatment[13]. Based on this work, OT has undergone clinical trial evaluation in adolescent RMS (NCT01858168). PARP inhibitors (PARPi), including olaparib, act primarily by inhibiting DNA homologous repair mechanisms and were demonstrated to be effective in *BRCA*-deficient cancers, including ovarian carcinoma and breast cancer[19,20]. Yet, some PARPi have an additional mechanism of action by which they inhibit tumor growth by locking PARP proteins on single-stranded DNA during cell division, causing replication fork collapse in a process known as "PARP-trapping"[21,22]. Strong PARP-trappers, when combined with DNA-damaging agents including temozolomide, induce cell cycle arrest and subsequently kill cancer cells that are not DNA-repair deficient, including RMS[13]. Despite the excitement surrounding the clinical deployment of OT, recent trials have also suggested that chronic OT treatment can result in therapy resistance in other tumor types[23–25]. For example, OT resistance can result from altered chromatin architecture induced by CYP1B1/H1.4 in ovarian cancer or by activating error-prone DNA polymerases during replication to repair therapy-induced DNA breaks in non-small cell lung cancer (NSCLC)[26]. Yet, to date, the mechanisms governing RMS resistance are largely unknown, and strategies to kill resistant RMS have yet to be developed. A detailed molecular understanding of therapy resistance in RMS will be important for developing effective therapies for the future, including those that kill VAC-resistant tumors, which is currently the largest clinical hurdle facing patients.

Here, we have developed OT resistant RMS models by treating xenografted mice for up to five cycles of therapy. Using a combination of next-generation sequencing, a targeted drug screen, and gold standard preclinical mouse xenograft studies, we show that chronic OT therapy can induce acquired resistance that is associated with rapid efflux of drugs from RMS cells. We find that a large fraction of OT and VAC resistant RMS activate the PIK3CA/AKT pathway to induce NRF2 mediated transcription of multiple multidrug-resistant ABC transporters. Preclinical modelling using xenografted tumors grown in immunodeficient mice demonstrated that inhibition of PI3Kα subunit with FDA-approved alpelisib potently re-sensitized RMS cells to killing by OT through suppressing expression of the ABC transporters and ultimately led to retention of drugs within the cell. Importantly, a large fraction of VAC-resistant RMS utilize this same molecular pathway to drive resistance and can be effectively killed by combination of OT and alpelisib. Together, our work defines a dominant resistance mechanism in RMS and has provided preclinical rationale for targeting the PIK3CA/AKT pathway as a counterstrategy to reverse therapy resistance.

## Results

### OT resistance develops in a faction of RMS models

OT therapy has demonstrated exceptional clinical efficacy in a range of cancer types[16–18,27,28]. However, recent clinical trial data has indicated that patients can acquire resistance to OT in other cancers[29]. To assess if this may also be the case in RMS, we engrafted *NOD.Cg-Prkdc^scid Il2rg^tm1Wjl/SzJ* (NSG) mice with human RMS and assessed the long-term responses to OT therapy using clinically-relevant oral dosing for up to five cycles of treatment in vivo [1 cycle = 5 days on/16 days off; *n* = 6 mice per treatment group; olaparib (daily 50 mg/kg) and temozolomide (daily 25 mg/kg), Fig. 1][13]. Patient-derived xenografts (PDXs) were

provided from the St. Jude Childhood Solid Tumor Network[30] and Memorial Sloan Kettering Cancer Center. Among the six RMS xenograft models tested, two fusion-positive PDXs had complete responses to OT therapy administered over five cycles and no animals developed therapy resistance (MAST118 (FP) and MSK82489 (FP), Fig. 1C). By contrast, the remaining models developed variable resistance across xenografted mice including fusion-positive Rh41 and fusion-negative RD, PDX MAST 139, and PDX MAST39 (FN; Fig. 1 and Supplementary Fig. 1). As may have been expected, cell proliferation (Ki67) and apoptosis (TUNEL) were not significantly different between OT resistant and control parental therapy-sensitive tumors (Fig. 1A and Supplementary Fig. 1). In total 20 out of 36 xenografted RMS tumors developed OT resistance (Fig. 1B, C).

To ask if OT resistance can be generalized to a wider cohort of models, we exposed an array of RMS cell lines to sustained and escalating doses of OT in vitro for up to 120 days (Supplementary Fig. 2, FN: SMS-CTR, JR-1, RMS176, 38IT, RMS559; FP: Rh30, Rh5, Rh3). As was seen in our in vivo modeling, two FN-RMS models were effectively killed by OT treatment (381T and RMS559), while the remaining six RMS models developed resistance over time as determined by growth under escalating doses of OT. Together, our in vivo xenograft modeling and in vitro cell culture work demonstrated that OT therapy effectively curbed tumor growth in all models tested, but was only able to fully kill tumor cells in four RMS models. Indeed, a majority of models acquired resistance following long-term, chronic OT treatment (*n* = 10).

### OT resistant RMS do not have common, shared acquired somatic mutations

Chronic OT treatment in NSCLC PDX models can induce high levels of genomic mutations[31]. Indeed, temozolomide is a well-known alkylating agent that is highly mutagenic and commonly leads to elevated point mutations in the genome[32]. Moreover, PARP1 inhibitors downregulate DNA homology repair responses, likely to potentiate temozolomide-induced DNA damage and mutagenesis[33,34]. Finally, RMS commonly mutationally activate or suppress signaling pathways critical for RMS initiation and progression, including PIK3CA/RAS/AKT and P53, respectively[2,3,5,35]. To assess if similar mechanisms were causing acquired OT resistance in RMS, we leveraged the MGH clinical-grade single base primer extension assay (SNaPshot) to access genomic mutations in 120 commonly mutated cancer oncogenes and tumor suppressors[36,37]. From this analysis, we failed to identify any recurrent gain/loss-of-function genomic mutations or copy-number variations that correlate with acquired OT resistance across the eight models analyzed, including in genes known to regulate the PIK3CA/AKT pathway (*PIK3CA, EGFR, HER2, FGFR4, IGFR1, PDGFRa, CMET,* and *ALK*; Supplementary Data 1)[38,39]. As may be expected given the mutagenic capacity of temozolomide, 60× whole genome DNA gene sequencing (WGS) identified elevated somatic single nucleotide variants (SNV), small insertions and deletions (INDEL), structural variants (SV) and copy-number variants (CNV) in four resistant tumors when compared to parental sensitive tumors (Fig. 2A and Supplementary Fig. 2). Most of these mutations were subclonal, reflecting heterogeneity of mutations induced across cells following OT treatment (Supplementary Fig. 3). Yet, this same analysis failed to identify any recurrent mutations, indicating that OT-induced mutations may merely be passenger events that do not directly confer resistance (Fig. 2A, B, Supplementary Fig. 3C, and Supplementary Data 2 and 3).

Given the lack of recurrent genetic abnormalities observed after developing OT resistance across our models, we posited that OT resistance might be regulated by non-genetic mechanisms that would be predicted to be stochastically acquired over time and subsequently lost following growth of tumors in the absence of therapy. Indeed, others have identified the existence of non-genetic, reversible therapy-persister cells in other cancer types[40–42]. To test

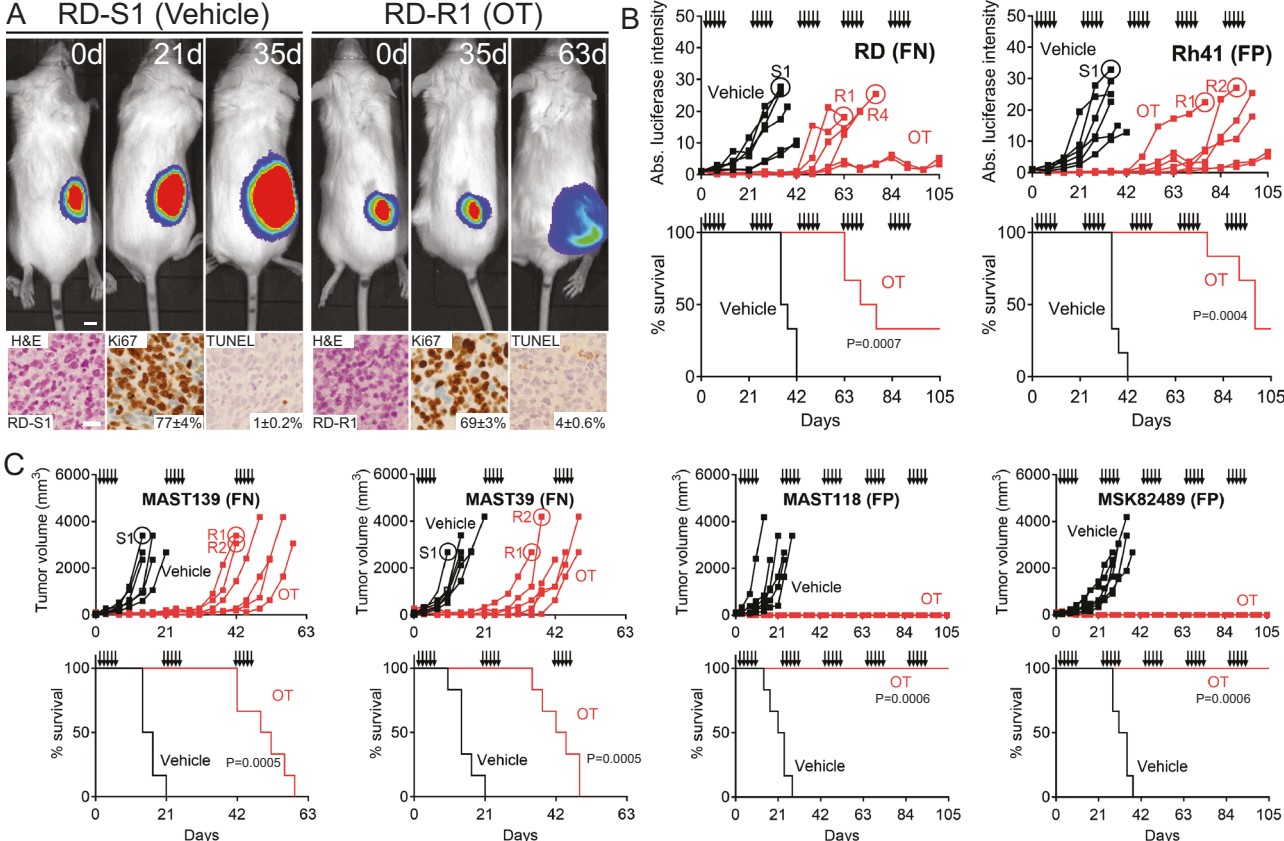

**Fig. 1 | Olaparib and temozolomide combination therapy inhibits human RMS xenograft growth, yet over half of animals develop therapy resistance over time. A** Representative images of NSG mice engrafted with FN-RMS RD cells (top) and histopathology analysis (bottom). Luciferase bioluminescent imaging was completed prior to drug administration (0 day) and after treatment as noted (**A**, top). For histological analysis, tumors from 3 different mice (biological replicates) were sectioned, and 3 image areas per tumor (technical replicates) were captured and analyzed. Representative hematoxylin and eosin-stained sections, IHC staining for Ki67 and TUNEL with average percent positive cells ± standard deviation (SD) noted (**A**, Bottom, *n* = 3 image areas/slide). **B**, **C** Quantification of RD and Rh41 RMS xenografted tumor growth assessed by luciferase imaging (**B**, top panels) or PDX models assessed by caliper measure in individual mice (**C**, top panels) and Kaplan–Meier Survival analysis (**B**, **C**, bottom panels). *N* = 6 mice per arm. Log-rank (Mantel–Cox) test was used to compare the significance. The five days of drug administration are denoted by black arrows in (**B**, **C**). RMS tumors that were harvested, dissociated, and used in subsequent studies noted by numbered circles. *P* < 0.05 was considered statistically significant., Log-rank Statistic. Scale bar equals 0.5 cm (**A**-top), 25 μm (**A**-bottom). Fusion-positive (FP) and fusion-negative (FN) RMS noted.

this hypothesis, we used Fluorescence-Activated Cell Sorting (FACS) to isolate single therapy resistant cells from four individually derived xenograft models and grew tumors from single cells (RD-R1, RD-R4, Rh41-R1, and Rh41-R2, Fig. 2C). We then analyzed two single-cell derived clones from each model for responses to therapy. As expected, all single-cell derived models were initially resistant to OT therapy at early passage. Yet, following long-term, serial passaging in the absence of drug for up to 44 passages (>120 days), most models reverted to a therapy-responsive cell state (*n* = 6 of 8 models, Fig. 2D, E and Supplementary Data 4). Because our experiments started with isolated, single therapy-resistant cells, these responses could not be accounted for by initial genetic heterogeneity within the culture and outgrowth of rare cell types that were predisposed to responding to OT. The loss of acquired resistance in serial passaged OT resistant cells indicates that therapy resistance is not likely genetically regulated.

We and others have recently identified cell state heterogeneity in human RMS using single-cell RNA (scRNA) sequencing and functional studies[43–46]. From this work, we uncovered a therapy-resistant cancer stem cell in FN-RMS[43,47]. Thus, we reasoned that our RMS-resistant models might have increased overall numbers of cancer stem cells. To address this possibility, we performed scRNA sequencing on parental therapy-sensitive and resistant tumors for each model (RD-R1, Rh41-R2, MAST39-R1, and MAST139-R1, Fig. 2F, G). Unexpectedly, scRNA sequencing did not identify significant cell state differences between OT resistant and sensitive tumors across the models examined (Fig. 2F). Yet, our analysis did uncover general transcriptional upregulation of known mediators of the AKT pathway, including *VEGFA* and *IGF2* the PDX-resistant cells (Fig. 2G and Supplementary Fig. 4)[48,49]. Other factors known to regulate the AKT pathway were also highly expressed in resistant tumors, including *PDGFA*, *CCL2*, *CTGF*, and *IGFBP2* (Fig. 2G, Supplementary Fig. 4, and Supplementary Data 8)[50–53]. Taken together, these findings suggest OT therapy resistance is not associated with elevation of a specific, transcriptionally defined RMS cell state and yet also suggests the possible link of therapy resistance with AKT pathway activity.

## A majority of OT resistant RMS upregulate the PIK3CA/AKT and MDR pathways
Given the lack of recurrent driver mutations detected in OT resistant RMS and the known roles for epigenetic transcriptomic regulation in driving drug-resistant cell states in other tumors[41,42,54,55], we hypothesized that there might yet still be a common signaling cascade responsible for driving OT resistance, including activation of the AKT pathway, as was suggested by scRNA sequencing studies. Indeed, others have shown that IGF1-inhibitor ganitumab prevents the

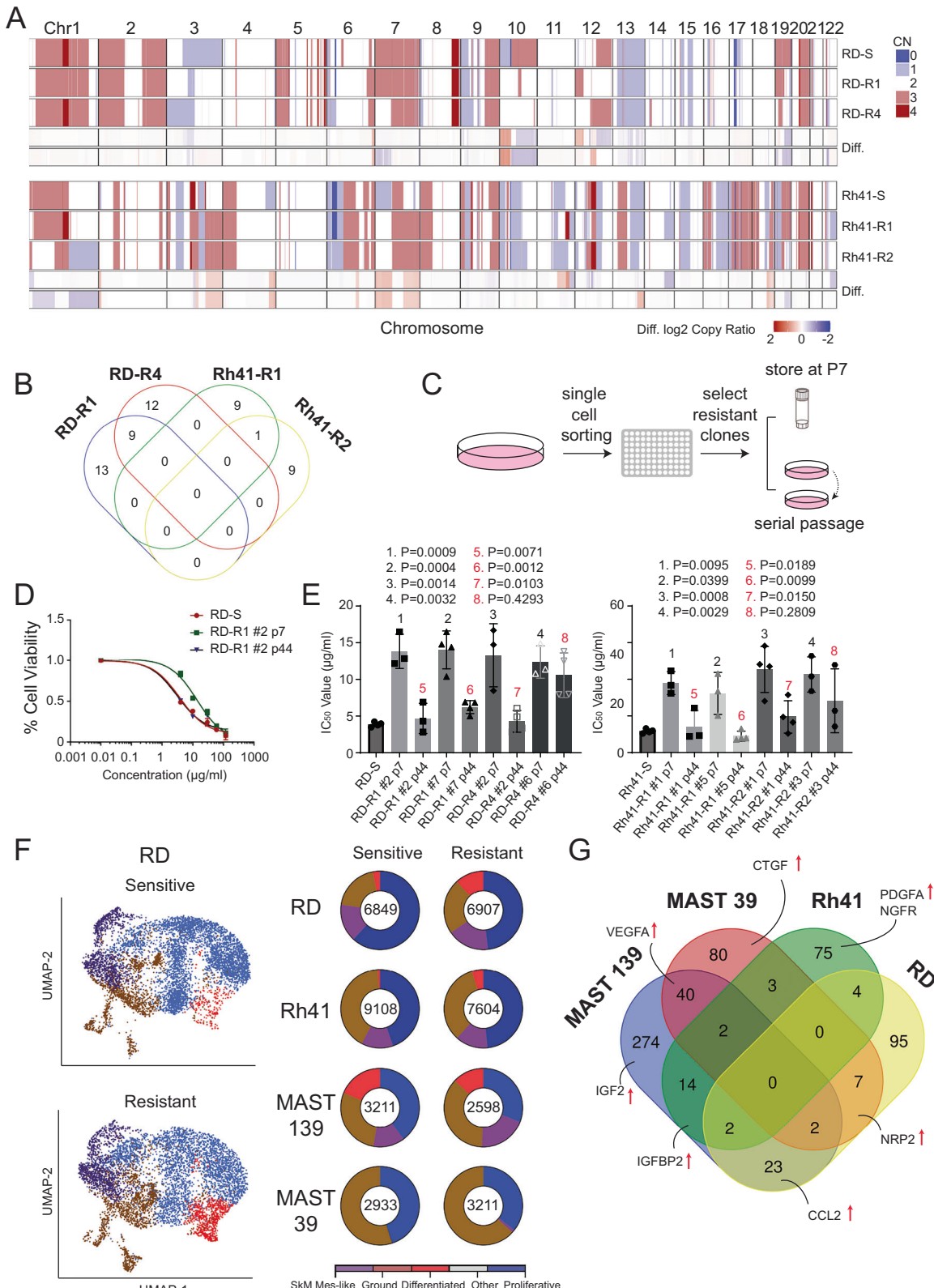

phosphorylation of AKT and synergizes with MEK inhibition to kill FN-RMS[56]. This therapeutic response was observed in models without mutational activation of the PI3K/PTEN axis. Moreover, high phosphorylation levels of AKT are associated with poor overall and disease-free survival in RMS[57], further suggesting a possible link of this pathway in elevating aggression and resistance. To clarify the molecular pathways that might confer OT resistance, we first performed immunohistochemistry staining of xenografted tumors using antibodies to known pathway regulators of RMS carcinogenesis and aggression, including EGFR, PTEN, PIK3CA, p-AKT, p-NRF2, and MDR1 (ABCB1)[3,5,9,44,58]. Of the proteins examined, PIK3CA, MDR1, p-AKT, p-NRF2, were increased in a majority OT resistant tumors when compared to therapy-sensitive parental tumors (Fig. 3A, B and Supplementary Fig. 5).

**Fig. 2 | OT resistance does not result from recurrent acquired somatic mutations or elevated numbers of known therapy-resistant RMS cell states.**
**A**, **B** Whole genome sequencing reveals few recurrent alterations between parental, sensitive, and resistant models. Genome-wide view of cumulative copy number variations (CNVs) present in RD (top panel) and Rh41 (bottom panel) models. Absolute copy number is provided to the right. Gains are indicated in red and loses in blue. Differences (Diff.) between parental and resistant clones are noted in the bottom panels (**A**). Venn diagrams representing the number of acquired genes variants (SNVs and INDELs) shared between the four resistant tumors in exon and splicing regions (**B**). **C–E** Therapy-resistant RMS models regain sensitivity to OT following serial passaging in the absence of drug. Schematic of experimental design created in BioRender. Mathavarajah, S. (2025) https://BioRender.com/7s4gaxy (**C**). Cell viability assessed by CellTiter-Glo for representative therapy-resistant and serially passaged clone treated with OT for 4 days (**D**, data obtained from three independent biological replicates (mean ± SD)). Quantification of IC$_{50}$ values for the six models that reverted to a therapy-responsive state after serial passaging (data shown for three independent experiments, mean/center noted by top of bar graph and error bars denote mean ± SD, **E**). Black lettering denotes differences in therapy responses to parental, sensitive lines (RD-S and Rh41-S) by ANOVA followed by Dunnett post hoc test. Red lettering denotes differences between early and late passaged cells from the same models by ANOVA followed by two-sided Student's T-test. Not significant (ns). $P < 0.05$ was considered statistically significant. **F**, **G** Single cell RNA sequencing reveals little change in the overall fractions of RMS cells based on molecularly defined cell states, but did identify genes that were upregulated within each model. TSNE plots for representative RMS model (**F**, left) and quantification of cell states with the number of cells analyzed shown within the center of each wheel chart (**F**, right). Venn diagram showing the overall number and selected genes upregulated in comparing parental, sensitive, and resistant models (**G**). Models analyzed were RD-S, RD-R1, Rh41-S, Rh41-R2, MAST39-S, MAST39-R1, MAST139-S, and MAST139-R1 (**F**, **G**).

To identify the dominant pathways that might drive OT resistance, we next dissociated OT resistant and therapy sensitive, parental RD and Rh41 xenografted tumors and cultured these cells in vitro with and without pathway-specific inhibitors that were predicted to be upregulated in resistant tumors (Fig. 3C). As expected, these cell models retained their resistance to OT in vitro after short passage number (Supplementary Fig. 6) and had the same signaling pathways upregulated when analyzed by Western blot (Fig. 3B). In total, we assessed 13 small molecule inhibitors for their ability to kill resistant RD and Rh41 tumors alone or in combination with OT (Fig. 3C). These experiments included FDA approved and investigational inhibitors of MAPK/ERK (Multi-tyrosine kinase—cabozantinib; MAPK—neratinib), EGFR (dacomitinib, osimertinib), PI3K/AKT (PI3Kα—alpelisib, PI3Kδ—duvelisib, PI3Kγ/δ—idealisib, mTOR1—everolimus, mTOR1—temsirolimus, mTOR2—vistusertib), and MDR inhibitors (tariquidar—pan-ABC transporter, zoquidar—ABCB1, dofequidar—ABCB1/ABCC1; Fig. 3C). Interestingly, PIK3CA/AKT pathway inhibitors robustly killed resistant tumors in the presence of OT and yet had only moderate impact on tumor cell killing in the parental, therapy-sensitive cells (Fig. 3C and Supplementary Data 5). These same pathway inhibitory drugs had little effect on overall growth as single agents in both therapy-sensitive parental models our resistant RMS (Fig. 3C).

Our immunohistochemistry experiments uncovered elevated MDR1 ABC transporter expression in resistant clones and chemical epistasis experiments showed that pan-ABC transporter inhibitors effectively killed resistant tumors only in the presence of OT and had limited effects on parental, therapy-sensitive cells (Fig. 3C and Supplementary Fig. 7). These results suggested that drug efflux may underly RMS resistance to OT therapy. To address this possibility, we next used a fluorophore-tagged olaparib analogue (PARPi-FL) to dynamically visualize the impact of drug uptake and retention in RMS cells[59]. Importantly, the PARPi-FL had similar drug properties in killing RMS cells as unlabeled olaparib, akin to that previously reported (Supplementary Fig. 8)[59]. Consistent with a role for upregulation of ABC transporters in driving OT resistant RMS, PARPi-FL was retained within parental, sensitive RD and Rh41 models, but not in resistant models (Fig. 3D, E). Yet, when co-treated with alpelisib or tariquidar pan-ABC transporter inhibitor, PARPi-FL was able to enter the cells and was retained within resistant cells (Fig. 3D, E). To further confirm elevated ABC transporter function in therapy-resistant models, we next performed the eFluxx-ID® Green assay that uses a hydrophobic, cell-permeant, non-fluorescent compound that fluoresces green after hydrolyzation by intracellular esterases and quantifies dye efflux by ABC transporters. As would be expected if ABC transporters are more active in resistance models, the probe was actively effluxed from cells and lower fluorescence was detected in all resistance models (Fig. 3G and Supplementary Fig. 9A). Further, treatment with MDR inhibitors increased probe retention in the resistant RMS cells, but not the sensitive clones (Supplementary Fig. 9A). More importantly, treatment with alpelisib alone led to inhibition of ABC transporter function in the resistant RMS models, resulting in retention and high level expression of eFluxx-ID® Green after 24 h of alpelisib treatment (Fig. 3F, G and Supplementary Fig. 9B, C). Treatment of tariquidar pan-ABC transporters inhibitor also re-sensitized the resistant RMS cells to OT treatment (Supplementary Fig. 7). Together, these data show that OT resistance is associated with elevated PIK3CA and ABC transporter activity in a large fraction of OT-resistant models in both FN- and FP-RMS.

To assess if the PIK3CA/AKT pathway directly confers therapy resistance, we lentivirally expressed activated PI3Kα (H1074R) or myristoylated (myr)-AKT1 in OT sensitive, parental RD and Rh41 cells (Fig. 4 and Supplementary Data 6)[60,61]. Constitutively active PI3Kα and myr-AKT increased the mRNA and protein expression of MDR1, MRP1, and BCRP drug efflux transporters (Fig. 4A, B, E, F) and induced OT resistance (Fig. 4C, G). Live-cell imaging of PARPi-FL confirmed that RD and Rh41 cells that had engineered activation of the PIK3CA/AKT pathway could efficiently efflux PARPi-FL (Fig. 4D, H). Together, these gain-of-function experiments confirm the role of the PIK3CA/AKT pathway in upregulating both ABC transporter expression and function, ultimately leading to OT resistance.

## The PIK3CA/AKT pathway controls ABC transporter expression through NRF2

Next, we asked which downstream effector molecule of the PIK3CA/AKT pathway is responsible for ABC transporter expression. Our IHC and Western blot studies demonstrated upregulation of p-NRF2 (Fig. 3A, B and Supplementary Fig. 5), which is known to regulate transcription of a wide array of ABC transporters in other cancers[62-64]. Using siRNA gene knockdown of *NRF2* in RD and Rh41 OT resistant tumor cells, we showed that *MDR1*, *MRP1*, and *BCRP* were significantly reduced at both the mRNA and protein levels in all models tested following *NRF2* knockdown (Fig. 5A, B and Supplementary Fig. 10A, B). Further, *NRF2* knock down led to robust retention of PARPi-FL in resistant RD and Rh41 tumors (Fig. 5C and Supplementary Fig. 10C) and reduced cell viability when exposed to OT (Fig. 5D and Supplementary Fig. 10D).

To further confirm the functional role of NRF2 in regulating ABC transporter expression, we next conducted chromatin immunoprecipitation analysis of NRF2, akin to studies reported for ABCC1 and ABCG2 in human lymphoid cells, breast cancer, lung cancer and prostate cancer cells[65-67]. Indeed, ChIP-qPCR analysis revealed highly elevated NRF2 binding to the promoters and enhancers of *MDR1*, *MRP1*, and *BCRP* in all resistant models tested (Fig. 5G and Supplementary Figs. 10G and 11). As would be expected if the PIK3CA pathway regulates NRF activity, treatment of resistant models with single agent PIK3CA inhibitor alpelisib reduced ABC transporter expression at both the RNA and protein level (Fig. 5E, F and Supplementary Fig. 10E, F); reduced NRF2 phosphorylation and overall

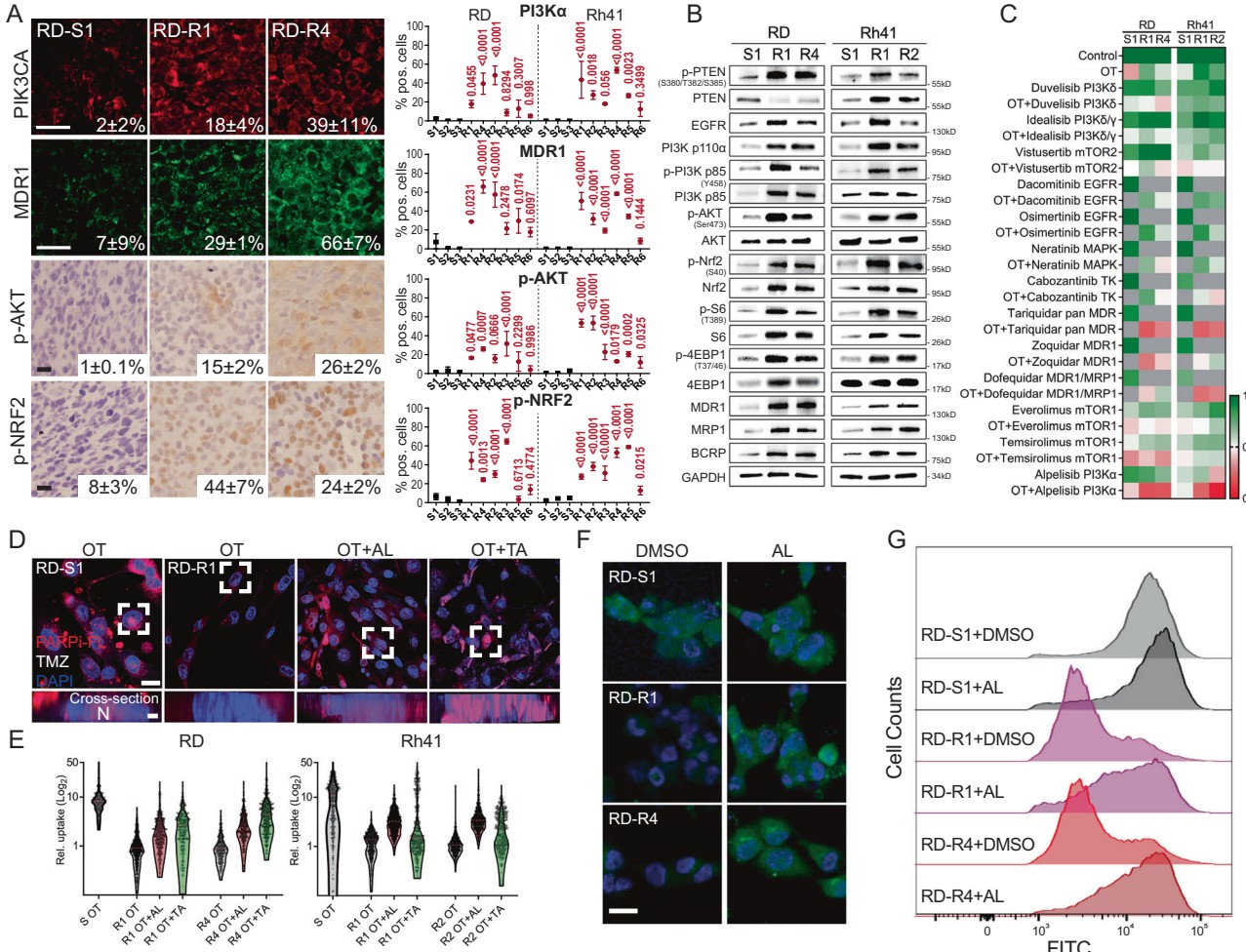

**Fig. 3 | OT resistant tumors activate the PIK3CA/AKT pathway, have elevated ABC transporter expression, and rapidly efflux drug from cells.**
**A** Histopathology of xenografted tumors. Representative RD parental sensitive (S1) and resistant models (R1, R4) showing co-immunofluorescence staining of PI3Kα (Alexa Fluor 546) and MDR1 (Alexa Fluor 488); IHC for p-AKT (Ser473) and p-NRF2 (Ser40; **A**, left). Quantification of percent positive cells within each model mean +/− ST noted on images or in graphical analysis (right). $N = 3$ in sensitive models and $n = 6$ in resistant models (biological replicates, RD and Rh41). Data are mean ± SD. **B** Western blot analysis of RD and Rh41 cell line grown after selection in xenografted mice. Data shown are from one representative experiment and was repeated three times using biological replicates with similar results. The lysates derive from the same experiment but were run on different gels. One gel was used for analysis of PI3K p110, Nrf2, p-Nrf2, AKT, p-AKT, p-4EBP1, S6, another for EGFR, PI3K p85, p-PI3K p85, MDR1, MRP1, BCRP, p-PTEN, PTEN, 4EBP1, and p-S6. Both gels were processed in parallel. **C** Quantification of cell viability following drug treatment in RD and Rh41 models. Cells treated for 4 days. Grey denotes comparisons that were not done. Scale (1.0 growth/viable and 0.0 dead). Data shown as mean from three independent biological replicates. **D** Representative confocal image of RD therapy

sensitive and resistant model following treatment with PARPi-FL (red) and temozolomide (TMZ) or in combination with alpelisib PIK3CAα inhibitor (OT + AL) or Tariquidar ABC transport inhibitor (OT + TA). Samples were counterstained with NucBlue. Z-stack image (top, **D**) and cell cross-section (bottom, **D**). **E** Quantification of relative PARPi-FL uptake in therapy sensitive and OT resistant RD and Rh41 cells subjected to OT, OT + alpelisib (AL), OT + tariquidar (TA). $n > 248$ cells per condition. Shown is a representative example from one of three independent, biological replicate experiments; similar results observed across all experiments (**D**, **E**). **F** Representative images and **G** flow cytometry analysis of RD parental sensitive (S1) and resistant cells (R1, R4) showing fluorescence eFluxx-ID® signals after treating with DMSO or alpelisib PIK3CAα inhibitor (AL). The fluorescence signal of the dye negatively correlates with the activity of the ABC transporters. Images are representative of three independent biological replicates (**F**, **G**). $P < 0.05$ was considered statistically significant. ANOVA followed by Dunnett post hoc test was used to make comparisons between sensitive and resistant clones (**A**). ANOVA followed by two-sided Student's $T$-test comparisons within each resistant model (**E**). Scale bar equals 25 μm (**A**), 10 μm (**D**, upper; **F**), and 2 μm (**D**, lower).

expression (Fig. 5E and Supplementary Fig. 10E), and reduced overall ChIP enrichment of NRF2 at these same promoter/enhancer sites (Fig. 5G and Supplementary Fig. 10G). Together, these experiments demonstrated that the PIK3CA pathway regulates ABC transporter transcription by activating NRF2 binding to genomic regulatory regions. Notably, serial passaged models that had re-gained OT therapy responsiveness over time also had lower PIK3CA and reduced NRF2 activity ($n = 6$ models, Supplementary Fig. 12). These data further support our hypothesis that the PIK3CA/AKT pathway regulates NRF2 transcriptional activity to induce expression of ABC transporters in OT resistant RMS cells.

## Activation of the PIK3CA/AKT pathway also drives therapy resistance to VAC

To assess if the same molecular mechanisms also drive VAC resistance, we next exposed RMS cells to sustained and escalating doses of VAC for up to 120 days and then created single-cell derived cultures using FACS (Supplementary Fig. 13A). As expected, each single cell derived model was resistant to VAC after early passage, with many coordinately upregulating the PIK3CA/NRF2/ABC transport pathway when assessed by Western blot analysis ($n = 6$ of 13 models, Fig. 6A). Four of these models could be re-sensitized to VAC chemotherapy-induced killing after co-treatment with

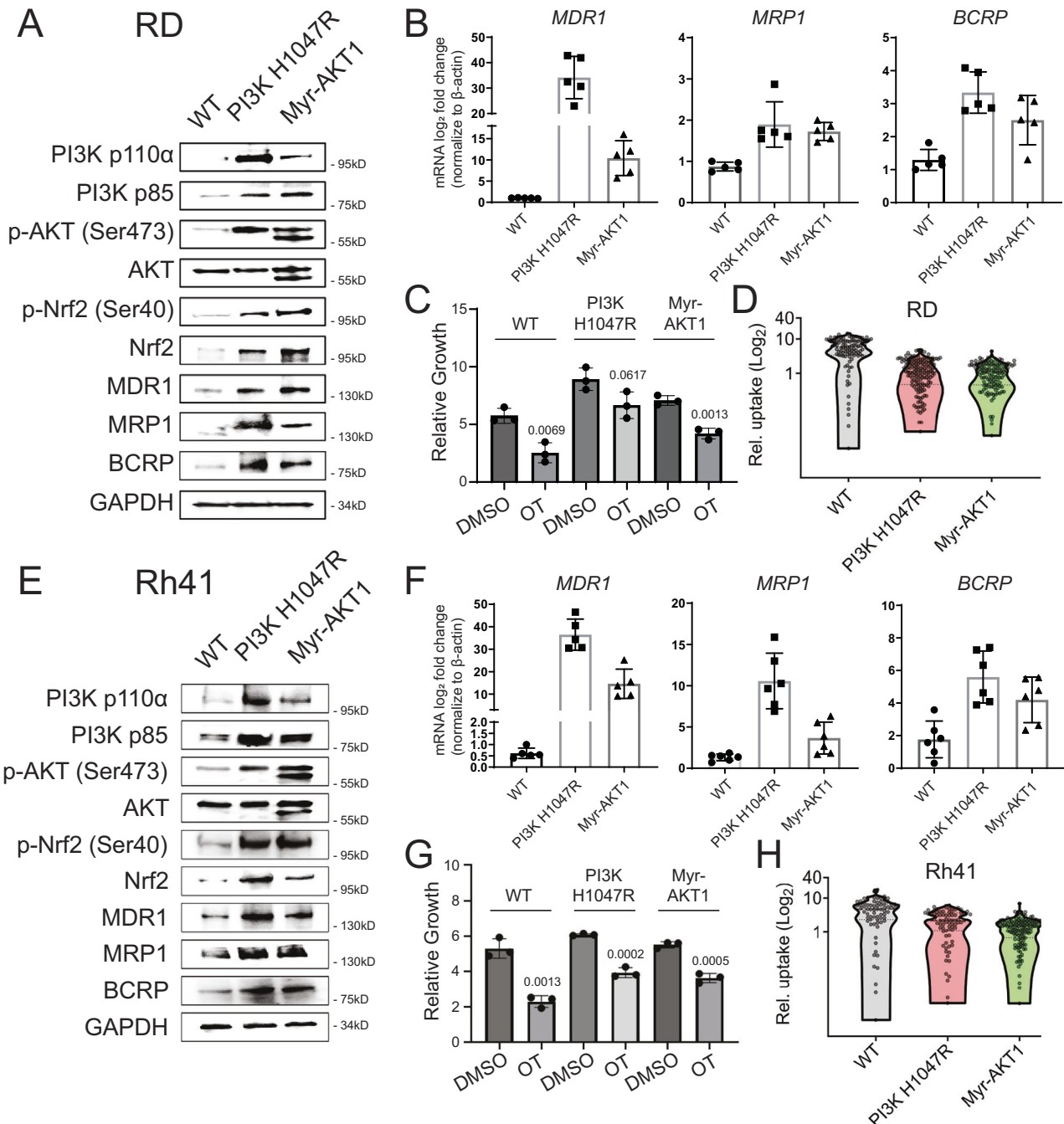

**Fig. 4 | Activation of the PIK3CA/AKT signaling pathway elevates ABC transporter expression and drug efflux.** RD (**A**–**D**) and Rh41 cells (**E**–**H**). **A**, **E** Western blot showing pathway activation in cells that lentivirally express activated PIK3CA (H1047R), myristoylated AKT1 (Myr-AKT1), or WT is empty vector. Shown are representative examples from one of three independent, biological replicate experiments. The lysates derive from the same experiment but were processed in parallel and run on different gels. For (**A**), one gel analyzed MDR1, Nrf2, and another analyzed PI3K p110, PI3K p85, MRP1, BCRP, P-Nrf2, AKT, P-AKT. While in (**E**), one gel analyzed MRP1, BCRP, a second analyzed PI3K p110, PI3K p85, Nrf2, P-Nrf2, AKT, P-AKT, and the third analyzed MDR1 (**E**). **B**, **F** qRT-PCR quantitation of ABC transporters. Normalized to GAPDH. Representative data showing mean ± SD from five technical replicates. Similar results were obtained from analysis of three biological replicates. **C**, **G** CellTiter-Glo analysis showing relative growth of cells after 5 days of treatment with DMSO or OT. *P* values denote differences in control cells before and after OT treatment. Data are mean ± SD from $n = 3$ biological replicates. **D**, **H** Quantification of drug uptake after 4 days of treatment of PARPi-FL with temozolomide. Number of cells per treatment group noted on the image. Shown is a representative example from one of three independent, biological replicate experiments; similar results observed across all experiments (**D**–**H**). Two-sided Student's *T*-test was used to compare cell killing by DMSO control or OT treatment within each model (**C** and **G**). $P < 0.05$ was considered statistically significant.

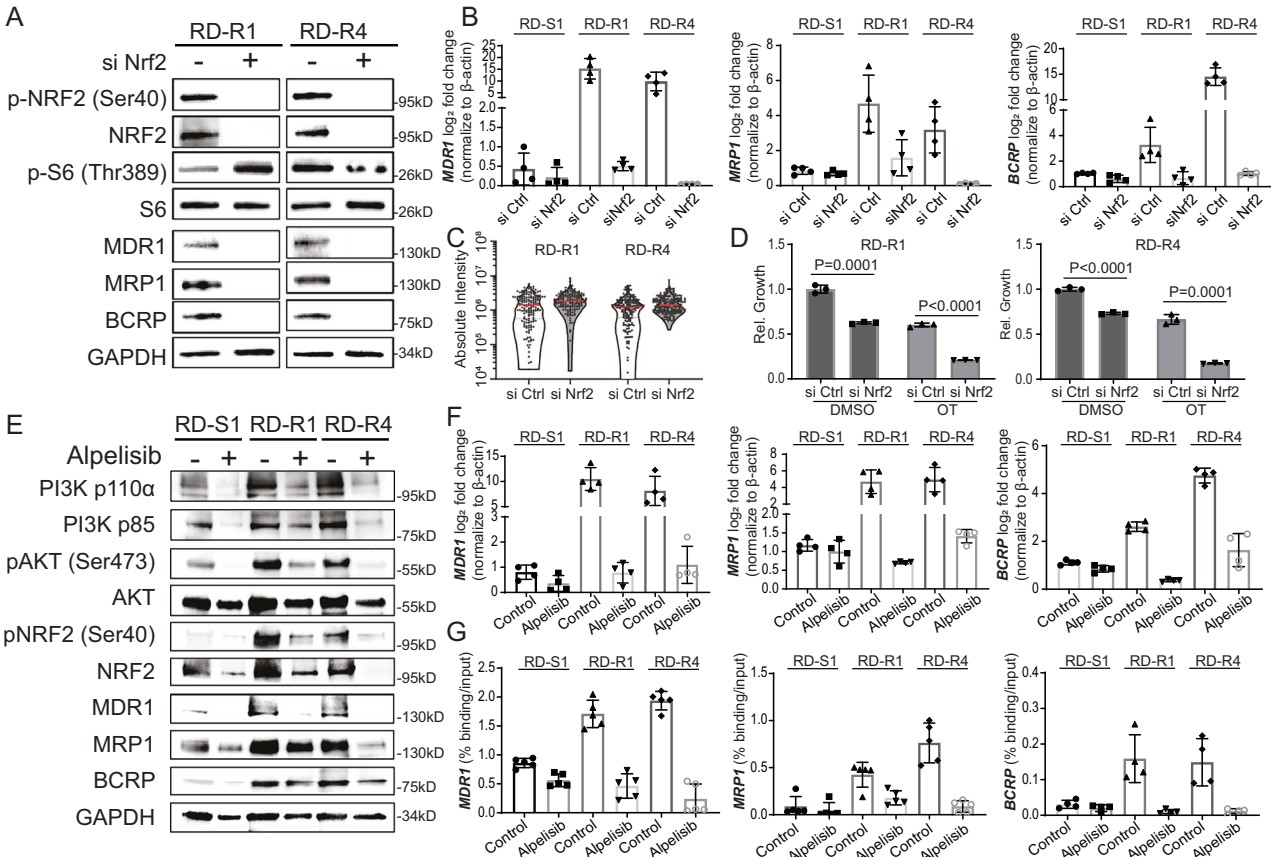

**Fig. 5 | NRF2 upregulates ABC transporter expression and function in OT-resistant RMS. A**–**D** siRNA knockdown of NRF2 results in reduced ABC transporter expression and function in resistant models. **A** Western blot analysis following 4 days of siRNA knockdown. Shown are representative examples from one of three independent, biological replicate experiments. The lysates derive from the same experiment but were processed in parallel and run on different gels. One gel analyzed MDR1, MRP1, BCRP, Nrf2, p-Nrf2, and a second analyzed S6, p-S6. **B** qRT-PCR showing that ABC transporters are transcriptionally upregulated in resistant clones and can be downregulated following siNRF2. *P* values in black denote comparison to RD-S control siRNA cells and red denotes comparison within each resistant model. Representative analysis showing mean ± SD from four technical replicates. Similar results were obtained from three independent experiments that used additional biological replicates. **C** Quantification of PARPi-FL within individual resistant models after 4 days of siRNA knock-down. $n \geq 153$ cells per condition. Shown is a representative example from one of three independent, biological replicate experiments; similar results observed across all experiments (**C**). **D** CellTiter-Glo analysis showing relative growth of cells after 7 days of DMSO or OT drug treatment in siControl or siNRF2 knockdown

cells. *P* values denote differences in siControl and siNRF2 in each drug treatment group. Data are mean ± SD from $n = 3$ biological replicates. **E**, **F** Alpelisib PIK3Cα inhibitor reduces ABC transporter expression. Western blot (**E**, representative example, completed three times using independent biological replicates) or qRT-PCR analysis (**F**, samples normalized to GAPDH) following 4 days of alpelisib treatment. qRT-PCR analysis shows mean ± SD from four technical replicates. Similar results were obtained from three independent experiments that used additional biological replicates. Shown in (**E**) are representative examples from one of three independent, biological replicate experiments. The lysates derive from the same experiment but were processed in parallel and run on different gels. One gel analyzed PI3K p110, PI3K p85, MDR1, MRP1, Nrf2, P-Nrf2, AKT, P-AKT, and a second analyzed BCRP (**E**). **G** ChIP-qPCR analysis of NRF2 occupancy on the ABCB1 (MDR1), ABCC1 (MRP1), and ABCG2(BCRP) promoters/enhancers following treatment with vehicle or alpelisib for 4 days. Data shows mean ± SD from four technical replicates. Similar results were obtained from two independent experiments that used additional biological replicates. *P* < 0.05 was considered statistically significant. Two-sided Student's *T*-test was used in (**C**, **D**).

alpelisib, suggesting that this same pathway can also drive VAC resistance (Fig. 6B and Supplementary Fig. 13B). Yet, in other models, VAC-resistant RMS could not be killed efficiently by co-treatment with alpelisib suggesting additional, redundant pathways likely induce VAC resistance beyond eliciting effects on the PIK3CA/NRF2/ABC transport pathway. All six VAC-resistant models that had PIK3CA/NRF2/ABC transport pathway activation were effectively killed by OT + alpelisib (Fig. 6B and Supplementary Fig. 13B). Together, our data shows that OT + alpelisib can kill VAC resistant tumors that activate the PIK3CA/NRF2/ABC transport pathway and identified potential biomarkers of resistant tumors that could be used to stratify patients into trials using VAC or OT + alpelisib in the future.

## Alpelisib re-sensitizes resistant RMS to OT in mouse xenograft models

We next assessed if the triple drug combination of alpelisib and OT could also effectively kill resistant RMS in vivo using the gold-standard NSG mouse xenograft model. Alpelisib + olaparib is currently in phase I clinical trial evaluation for breast cancer and ovarian carcinoma (NCT01858168). Using similar dosing schedule and concentration of alpelisib as in that trial and used in previous mouse xenograft studies, we engrafted $1 \times 10^6$ OT resistant PDX MAST139 or MAST39 subcutaneously into NSG mice and treated mice[68–70]. After engrafted tumors reached a volume of 200–300 mm³, mice were orally gavaged with either vehicle control (1% carbox-ymethylcellulose), single agent alpelisib (50 mg/kg), combination of

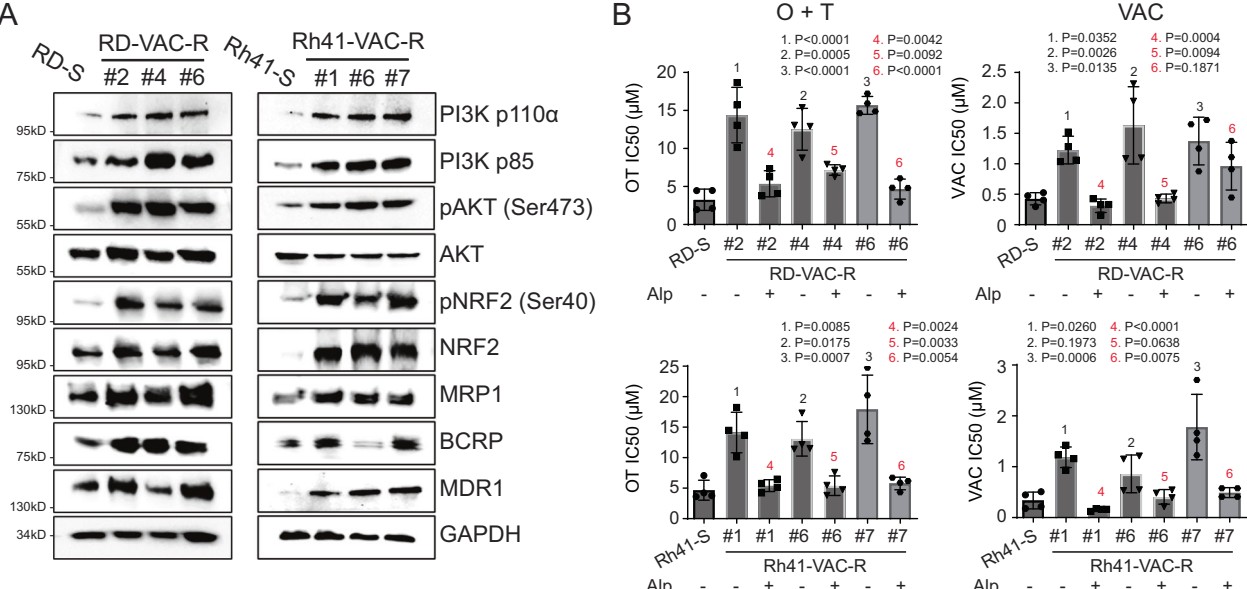

**Fig. 6 | A large fraction of RMS activate the PIK3CA/AKT pathway to drive therapy resistance to VAC. A** Western blot for representative VAC resistant RD and Rh41 clones. Shown are representative examples from one of three independent, biological replicate experiments. The lysates derive from the same experiment were processed in parallel and run on different gels. One gel analyzed RD samples, and a second analyzed Rh41 samples. **B** Quantification of $IC_{50}$ values for each VAC resistant RD and Rh41 model. Data are mean from four independent biological replicates ± SD. *P* values in black denote differences in comparison to parental, sensitive lines (RD-S or Rh41-S, one-way ANOVA followed by two-sided Dunnett test). *P* values in red denote differences between individual clones treated with either OT or VAC and those co-treated with alpelisib (one-way ANOVA followed by two-sided Student's *T*-test). *P* < 0.05 was considered statistically significant.

O (50 mg/kg) and T (25 mg/kg) or triple combination of OT and alpelisib (50 mg/kg, *n* = 6 mice analyzed per arm). We found that the triple drug combination of OT and alpelisib significantly suppressed resistant tumor growth as well as prolonged animal survival in the PDX models analyzed (Fig. 7A, B). Body weight of animals treated with OT and alpelisib showed no significant differences as compared to control, alpelisib or OT treated animals, with no additional observable side effects (Fig. 7C). Importantly, net tumor weight significantly decreased after treatment with OT + alpelisib but not in other treatment groups (Fig. 7D). Tumor suppressive effects were further validated with antibody immunohistochemistry staining of Ki67 to assess proliferation and TUNEL to analyze apoptotic responses (Fig. 7E). Importantly, alpelisib, with or without OT, robustly inhibited PIK3CA/AKT pathway activity, as denoted by a significantly decreased PIK3CA, p-AKT and p-NRF2 expression observed from immunostaining (Fig. 7E). Lastly, lowered PIK3CA/AKT pathway activity corresponded with diminished MDR1 expression in vivo (Fig. 7E). Together, we demonstrated that the combination of alpelisib + OT is tolerable in a NSG mice xenograft RMS PDXs, significantly decreased PIK3CA/AKT pathway activity and p-NRF2, and ultimately inhibited MDR mechanisms to decreased tumor volume and subsequently prolonged survival.

## Discussion

Our work has uncovered a common resistance mechanism where therapy-resistant RMS upregulate the PIK3CA/AKT pathway. Our finds are similar in some regards to previous work showing that P110α and P110β positively regulate ABCB1-mediated multi-drug resistance in human epidermoid carcinoma and NSCLC[71,72], yet also uncovered roles for the PIK3CA/AKT pathway in regulating NRF2-mediated transcription of ABC transporters[72]. Curiously, upregulation of the PIK3CA/AKT pathway is not driven by recurrent genomic mutations or changes in intratumorally cell state heterogeneity, but rather transcriptionally through regulators of the pathway. Subsequent activation of PIK3CA/AKT results in NRF2-dependent transcriptional activation of multiple

ABC transporters that are well-known to rapidly efflux drugs from the cell to protect them from drug-induced DNA damage, cell cycle arrest, and apoptosis[73,74]. Elevated ABC transporters protein expression has been previously reported in RMS patients after therapy[75], yet the common upstream modulators of their expression and function were largely unknown. Our work also identified that NRF2 is the dominant transcriptional regulator of ABC transporters in OT resistant RMS, including *ABCB1, ABCC1*, and *ABCG2* that are known drug efflux pumps for olaparib and temozolomide. NRF2 is best known for regulating redox metabolism, autophagy, and unfolded protein responses[76], and AKT activity regulates NRF2 expression in part through the GSK3β/β-TrCP axis[77,78]. In cancer, NRF2 is well-known for regulating oxidative stress, mostly in the context of KEAP1-mutant cancers[79]. Rather, our work identified NRF2 as a dominant regulator of drug efflux in therapy-resistant RMS by transcriptionally modulating a wide array of ABC transporter gene expression. Indeed, NRF2 has lesser-known roles in binding to promoter regions and upregulating expression of the *ABCB1* and *ABCG2* drug efflux transporters directly impacting drug retention within cancer cells[62,80].

Based on our discovery of a dominant underlying mechanism of OT resistance in our xenograft RMS models, we also credentialed PIK3CA/AKT pathway inhibition as a preclinical therapy that re-sensitizes RMS to chemotherapy-induced tumor cell killing by co-treating them with alpelisib PIK3CA inhibitor. Importantly, safety data exists for use of alpelisib in children and has recently received FDA approval for PIK3CA-related overgrowth spectrum[81,82], suggesting a possible path forward for future clinical translation. This OT + alpelisib drug combination was well-tolerated and effectively killed both OT and VAC-resistant RMS, raising hopes that this combination might move into clinical evaluation in the future in the setting OT resistance and/or as a possible salvage combination treatment for VAC-refractory RMS. Importantly, our work also identified that VAC resistance is mediated in part by upregulation of the same PIK3CA/NRF2/ABC transport pathway and these tumors can be effectively killed by OT + alpelisib. Indeed, high phosphorylation levels of AKT and mTOR are

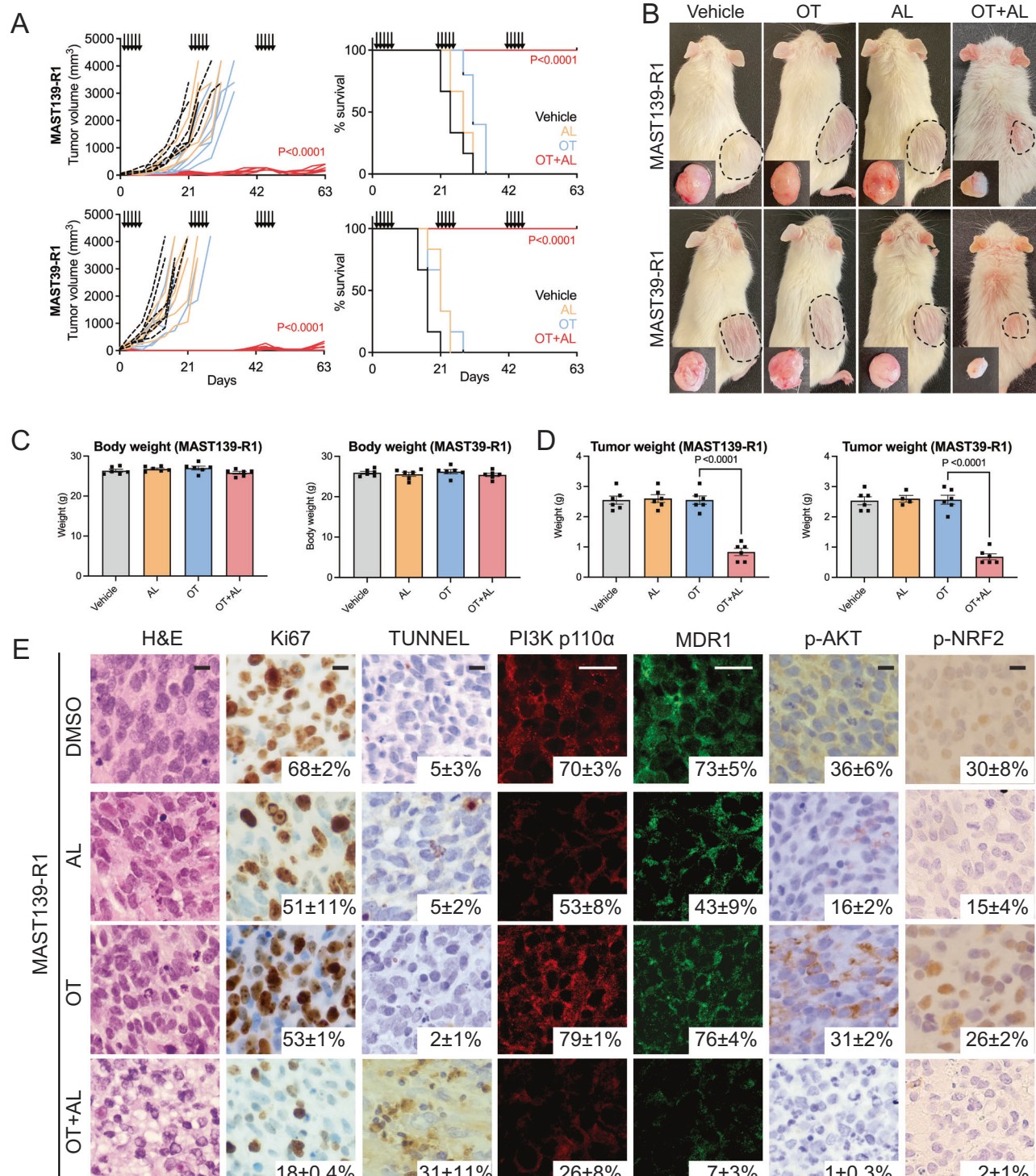

**Fig. 7 | Alpelisib PIK3CA inhibitor re-sensitizes resistant PDX models to OT combination therapy in mouse xenografts. A** Quantification of PDX tumor growth in individual mice (**A**, left, Fisher exact test) and juxtaposed to Kaplan–Meier survival analysis (right). *N* = 6 mice per arm. Log-rank (Mantel–Cox) test was used to compare the significance between vehicle control and OT + AL treated mice. Resistant models FN-MAST139 (R1, top) and FN-MAST39 PDX (R1, bottom). NSG mice administered 5 consecutive days of either vehicle control, OT, alpelisib (AL) or OT + AL (three dosing periods denoted by arrows). **B** Representative images of whole mouse or extracted tumor. **C** Body weight after tumor extraction. **D** Tumor weight at time of mouse sacrifice. Quantification used ANOVA followed by two-sided Student's *T*-test. All data are presented with center value being the mean ± SD (**C**, **D**). **E** Histopathology of representative MAST139-R1 tumor. Hematoxylin and eosin staining (H&E), IHC for Ki67, TUNEL, IF staining of PI3Kα (Alexa Fluor 546) and MDR1 (Alexa Fluor 488), p-AKT (Ser473), and p-NRF2 (Ser40, left to right). Quantification denotes the mean percent positive area +/− STD across three animals (three image planes quantified per animal, *n* = 3 biological replicates), *P* < 0.05 was considered statistically significant. Scale bars equal 1 cm (**B**) and 25 μM (**E**).

associated with poor overall and disease-free survival in RMS, especially in the context of VAC resistance[57]. ABC transporters are also upregulated in aggressive, therapy-resistant, and relapse RMS[83,84]. Our findings provide rationale for exploring the deployment of OT + alpelisib in the context of tumors that progress on current chemotherapy regimens and may provide opportunities to treat refractory disease in the future. Our work also suggests that activation of the PIK3CA/NRF2/ABC transport pathway, which can be assessed using standard IHC on paraffin-embedded sections, might be useful biomarkers for stratifying patients into clinical trials that use combination of alpelisib with either VAC or OT or other salvage therapies commonly used in the refractory/relapse setting. Moreover, PIK3CA pathway activity could be assessed after some duration of VAC chemotherapy and patients with elevated pathway activity treated in combination with alpelisib. Should alpelisib + OT have unexpectedly adverse side effects in future clinical studies, additional clinically available PIK3CA inhibitors have been tested in children, including the dual PI3K/mTOR inhibitor samotolisib that was evaluated in the Pediatric Combo-MATCH (LY3023414) or copanlisib.

We did not observe recurrent mutations in known oncogenes or tumor suppressors after the emergence of OT resistance. Rather, we uncovered that the PIK3CA/AKT pathway was stochastically activated after therapy, likely by yet unknown epigenetic or non-genetic changes. Activation of the PIK3CA/AKT pathway then turns on NRF2-dependent transcriptional activation of multiple ABC transporters leading to the rapid efflux drugs from cells. We posit that this may result from stochastic drift within tumor cells, where molecular pathways are sampled when therapy is applied and ultimately selected based on the activation of pathways that render them resistant to therapy. In our case, a large fraction of OT-resistant RMS upregulate the PIK3CA/AKT pathway. Lack of recurrent upstream driver mutations may also suggest pathway activation could result from well-known feed-forward loops that regulate the PIK3CA/AKT pathway[85–89]. Our observation in OT resistant RMS tumors is largely in line with a previously reported transient and reversible subpopulation of drug-tolerant lung cancer cells that are regulated epigenetically by histone demethylation via the RBP2/KDM5A/Jarid1A axis[41]. Further, cell lineage tracing experiments have demonstrated the existence of reversible therapy-persister cells driven by non-genetic mechanisms[40]. Together, our results raise the interesting and provocative idea that therapy resistance in RMS may be commonly regulated by non-genetic mechanisms and cells can transit from therapy-sensitive to resistant states and vice versa based on chemotherapy pressure, likely accounting for why few genetic mutations have been found in relapse and refractory disease. This also supports the notion of using therapeutic targeting strategies that incorporate drug holidays to minimize toxicity to patients while also allowing tumor cells to revert to therapy-sensitive states where subsequent cycles of therapy will have maximal tumor cell killing effects.

## Methods

### Animal welfare assurances and husbandry
Mouse studies were approved by the Massachusetts General Hospital subcommittee on research animal care under #2013N000038 and human studies under institutional review board protocol #2009-P-002756. Mice were housed in the MGH CCM BCL2 mouse facility within the CNY149 facility with temperature of 70 °F (range of 65°–75°F), humidity at 30%–70% RH, and lighting cycle 7:00 am ON-7:00 pm OFF. Mice were anesthetized with 5% isoflurane for 10 min and then euthanized by aortic exsanguination.

### Human cell lines, PDXs, and authentication
Cell lines used in this study are RD (RRID:CVCL_1649), Rh41 (RRID:CVCL_2176), SMS-CTR (RRID:CVCL_A770), JR-1 (RRID:CVCL_J063), 381T (RRID:CVCL_1751), RMS176 (RRID:CVCL_QW87), RMS559

(RRID:CVCL_S640), Rh30 (RRID:CVCL_0041), Rh5 (RRID:CVCL_C358), Rh3 (RRID:CVCL_L415). PDXs were provided by St. Jude Children's Research Hospital and Memorial-Sloan Kettering Cancer Center. These PDX models were created from tumors of consented patients under IRB approval and shared with MGH under MTA as described[43] (human IRB protocol #2009P002756, PDXs used under this protocol are denoted by prefix MAST or MSK). All human cell lines and PDXs were authenticated by small tandem repeat profiling using the Whatman Flinders Technology Associates sample collection kit (ATCC) as described[9].

Cell lines were cultured according to the recommendations of ATCC's, supplemented with 10% FBS (Atlanta Biologicals) and 1% penicillin/streptomycin/glutamine (PSG) (Life Tech), and grown at 5% $CO_2$ at 37 °C. PDX cells were cultured in RPMI (Gibco) supplemented with 10% FBS (Atlanta Biologicals) and 1% PSG. Adherent cells were dissociated using 0.25% trypsin for 2 min.

### Plasmids, siRNA, and Lentiviral transfection
Plasmids, lentiviral, and siRNA work was approved by Partners IBC under protocol #2013B000039. Lentiviral vectors were made as previously published[9]. pLenti-CMV-GFP-puro (Addgene #17448), pHAGE-PIK3CA-H1047R (Addgene, #116500), pHRIG-Akt1 (Addgene, #53583) were purchased from Addgene. Vectors were packaged using 293T cells. 2 µg pCMV-dR8.91 and 0.2 µg pVSV-g were transfected along with Trans-LTI reagent (Life Technologies) based on the manufacturer's recommendations. RMS were infected with 1:1 volume viral particle to DMEM/10%FBS media with 1:1000 Polybrene (EMD Millipore) for 24 h. Cells were grown in DMEM media supplemented with 10% FBS and selected with puromycin after 48 h.

siRNA transfections were performed using DharmaFECT Transfection Reagents-siRNA transfection (Horizon Discovery). 2 nmol ON-TARGETplus Human NFE2L2 siRNA (Gene ID 6198) were used to knockdown expression of NRF2. Synthetic siRNA were suspended in RNase-free solution at 100 µM. $7.5 \times 10^4$ cells were plated into 6-well plate, cultured for 24 h before siRNA transfections according to manufacturer's recommendation.

### in vitro cell culture assays
For in vitro drug assays, RMS cells were plated into 96 well plates at a density of 2500 cells/well in a volume of 100 µl. Drugs were prepared in 100 µl medium and added to the cells on the next day. After 4 days of incubation, cell cycle and proliferation were assessed by luminescence or flow cytometry using the CellTiter-Glo® Luminescent Cell Viability Assay (Promega, Madison, WI) or Click-It EdU Kit (Life Technologies, Carlsbad, CA), respectively. Cell apoptosis was measured using luminescence or flow cytometry using the Caspase-Glo® 3/7 Assay (Promega, Madison, WI) and Annexin V Apoptosis Detection Kit APC (eBioscience, San Diego, CA) according to manufacturer's protocol. DMSO was used as negative control. The assay was performed across three biological triplicates.

For serial passaging experiments, low-passage therapy-resistant cells were grown in vitro, expanded, and FACS sorted as single cells into 96 plates supplemented with media. Two single-cell derived clones from each resistant model were serial passaged and replated every 3–5 days. After >30 serial replating and passaging, cells were compared with its low passage parental clone based on cell viability and protein expression as noted.

For VAC treatment, Vincristine sulfate (V8388-1MG, Sigma), Actinomycin D (A1410-5MG, Sigma), and Cyclophosphamide monohydrate (93813–100MG, Sigma) were dissolved in PBS for 96 mM VAC stock (0.061 mM Vincristine: 0.065 mM Actinomycin D: 96 mM Cyclophosphamide). 96 mM VAC stock solution was then diluted to the appropriate working concentration.

To access PARPi-FL uptake in vitro, tumor cells were exposed to fluorescent Olaparib (PARPi-FL, 5 µM) and temozolomide (10 µM)

for 4 days. Each group also had one plate with no treatment as a control. Cells were stained with NucBlue Live ReadyProbes Reagent (Hoechst 33342) and imaged using confocal microscope (Zeiss LSM710 inverted microscope) at the end of experiment. Maximum projection images were created from 30-micron stacks (5 µm per slice with 6 slices) using a 40× objective (NA, 0.45, total 400× magnification). Cell nuclei and PARPi-FL uptake were imaged using the 405 nm laser (emission = 350–470 nm, DAPI) and 546 nm laser (emission = 575–703 nm, PARPi-FL). PARPi-FL relative uptake was quantified by subtracting the average 546 nm fluorescence intensity of control samples from the drug treated samples, then taking log2 fold change. This analysis uses surface function in IMARIS.

## Protein extraction and Western blot analysis

Total protein were obtained from cell lines by lysis in 2% SDS buffer that contained protease inhibitors (Santa Cruz Biotechnology). Samples were then boiled, vortexed, and homogenized. 20–50 µg of protein was loaded onto 4%–20% Mini-Protean TGX gels (Biorad) and transferred to PVDF membranes. Western blot analysis was performed as previously described[25]. Primary antibodies used included PTEN (CST, 9559), p-PTEN (Ser380/Thr382/383) (CST, 9551), EGFR (abcam, ab52894), PI3K p85 (CST, 4292), p-PI3K p85 (Tyr458) (CST, 17466), AKT (CST, 4691), p-AKT (Ser473) (CST, 4060), NRF2 (abcam, ab62352), p-NRF2 (Ser40) (abcam, ab76026), S6 (CST, 2317), p-S6 (Ser235/236) (CST, 81736), 4EBP1 (CST, 9644), p-4EBP1 (Thr37/46) (CST, 2855), MDR1/ABCB1 (CST, 13342), MRP1/ABCC1 (CST, 72202), BCRP/ABCG2 (CST, 42078), GAPDH (CST, 5174). Secondary antibodies used included anti-mouse IgG, HRP-linked Antibody (CST, 7076), anti-rabbit IgG, HRP-linked Antibody (CST, 7074).

## eFluxx-ID® dye efflux assay for multidrug resistance

eFluxx-ID® Green MDR assay kit (ENZ-51029, Enzo Life Sciences (ELS) AG Lausen, Switzerland) was used for functional detection of ABC transporter proteins (MDR1, MRP1, and BCRP). Briefly, $1.5 \times 10^6$ cells (RD-S1, RD-R1, RD-R4, Rh41-S1, Rh41-R1, Rh41-R2) were incubated with Green detection reagent with and without specific inhibitors of MDR1 (Verapamil), MRP-1 (MK-571), and BCRP (Noviobiocin) according to the manufacturer's protocol for 45 min in 37 °C. Alpelisib treatment (5 µM) was performed 24 prior to the incubation with the eFluxx-ID® probe. Cells were imaged using confocal microscopy (Zeiss LSM710 inverted microscope) with 20× objective employing 488 nm laser or suspended in cold PBS for flow cytometry analysis.

## Gel shift assays (EMSA) and ChIP-qPCR

To test the binding efficiency of NRF2 to the promoters of ABC transporter genes, we performed ChIP-qPCR in RD and Rh41 cells. Cells treated with or without alpelisib were prepared with SimpleChIP® Plus Enzymatic Chromatin IP Kit (CST #9005) and ChIP performed using the IgG control and NRF2 antibodies, respectively. For ABCC1 and ABCG2 genes, we used previously published primers[90] For ABCB1, we tested 8 predicted binding sites of NRF2 in the ABCB1 promoter region the comprise the predicted minimal ARE enhancer sequence (g) TGACnnnGC[91] and identified one binding site that was highly enriched in ChIP DNA fragment of NRF2 (Supplementary Fig. 11). RD and Rh41 cells were harvested after 96 h of treatment with DMSO or alpelisib, and cross-linked at room temperature with 1% formaldehyde for 15 min. Chromatin was fragmented using sonication, ranging in size from 200 to 800 bp. Protein-DNA complexes were precipitated using anti-NRF2 Ab (Abcam, ab62352) or anti-IgG Ab at 4 °C overnight. Ab-protein-DNA complexes were then isolated using protein beads, eluted, and reverse cross-linked by incubation at 65 °C with 200 mmol/L NaCl. Quantification used PCR and comparison of input or ChIPed DNA using the PowerUp SYBR Green Master Mix on LightCycler® 480 qPCR instrument (Roche). For each gene, enrichments were normalized to the respective input control. The sequences of the primers are listed in Supplementary Data 7.

To further confirm the binding of NRF2 to the ABCB1 promoter, we performed Electrophoretic Mobility-Shift Assay (EMSA). NRF2 recombinant protein from OriGene (TP760529), the Electrophoretic Mobility-Shift Assay (EMSA) Kit from Invitrogen (E33075) were used to detect the DNA shift after protein binding. Protein-DNA samples were incubated at room temperature in 1× binding buffer for 30 min and then run on a 6.5% TBE nondenaturing polyacrylamide gel in 0.5× TBE buffer. Gels were then stained with SYBR® Green for 30 min to stain for DNA and imaged on a transilluminator to monitor for the shifted DNA bands.

## Mouse xenograft studies

Human RMS tumor cells were embedded into Matrigel at a 1:1 ratio at a final concentration of $1 \times 10^7$ cells/ml and injected subcutaneously into flanks of 8-week-old female, NOD.Cg-Prkdcscid Il2rgtm1Wjl/SzJ (NSG) mice (Charles River Laboratories, 100 µl per mice, $1 \times 10^6$ cells per mice). Mice were followed until tumor size reached a volume of 200–300 mm³. Mice were orally gavaged with vehicle control (1% carboxymethylcellulose or 10% 2-hydroxypropyl-β-cyclodextrin/PBS) or olaparib (50 mg/kg) and temozolomide (25 mg/kg). Olaparib was administered twice daily, 8 h apart and temozolomide administered once daily at 4 h after the first olaparib treatment. Triple drug combination of OT with alpelisib was assessed for efficacy in killing OT-resistant tumors. Animals were orally gavaged with either vehicle control (1% carboxymethylcellulose), single agent alpelisib (50 mg/kg), combination of olaparib (50 mg/kg) and temozolomide (25 mg/kg) or triple combination of olaparib (50 mg/kg), temozolomide (25 mg/kg), and alpelisib (50 mg/kg). Doses administered for Olaparib, temozolomide, and Alpelisib were selected based on previous published xenograft and clinical trial studies[13,81,82,92,93]. Specifically, alpelisib was either administered once a day in the single-agent group or premixed with temozolomide and administered once a day 4 h after the first olaparib administration in triple combination therapy. In all experiments, animals were treated for five days, allowed to rest for 16 days and this cycle repeated. Disease progression was assessed twice per week using the IVIS imaging system to measure luciferase bioluminescence in cell line xenograft models and by manual palpation and caliper measurements in PDX models[9]. Maximal tumor size was 20 mm diameter and was not exceeded in our work. No animals were sacrificed due to exceeding humane endpoints defined in our animal protocol. Humane endpoints include: weight loss greater than 15% of body weight, lack of movement or lethargy/weakness causing inability to eat or drink water, signs of significant pain and/or distress, labored breathing, or skin lesions. Mice were humanely euthanized for tissue harvest before reaching maximal tumor size for subcutaneous xenografts or when no tumor detected after treatment cycles. Tumor harvest was performed as described previously[43].

## Histology, IHC, IF, and quantification

Engrafted tumors were extracted, fixed in 4% paraformaldehyde, embedded in paraffin, and sectioned (5 mm thickness). Sections were stained by hematoxylin and eosin (H&E) or immunohistochemistry (IHC) and immunofluorescence (IF). For IHC or IF staining, the primary antibodies were rabbit monoclonal anti-Ki67 (Abcam), TUNEL (Thermo Fisher), anti-NRF2 (phospho S40) (abcam 76026), anti-pAKT (Ser473) (CST, 4060), anti-PI3K p110α (CST, 4249), and mouse monoclonal anti-MDR1 (Sigma, MAB4334), 1:200 dilution in blocking buffer, 100ul per slide. Secondary antibodies included Biotinylated Goat Anti-rabbit IgG antibody (Vectorlabs) and Biotinylated Horse Anti-mouse antibody (Vectorlabs), 1:500 dilution in blocking buffer, 100 µL per slide. Antibody protein detection was completed using Vectastain ABC Kit (Vectorlabs). For IF staining, secondary antibodies were Goat anti-Rabbit IgG (H + L) Alexa Fluor 546 (Invitrogen, A-11035)

and Donkey anti-Mouse IgG (H + L) Alexa Fluor Plus 488 (Invitrogen, A32766), 1:1000 dilution in blocking buffer, 100 µl per well. Cell proliferation (based on Ki67) and apoptosis (based on TUNEL) were quantified. IHC stained sections were quantified as number of cells per unit area, as previously described[9]. IF stained sections were analyzed by taking an initial threshold and quantified using "3D Objects Counter" plug-in in ImageJ and shown as numbers of positive cells per unit area.

### scRNA-seq, Snapshot analysis, and whole genome sequencing

For single-cell RNA sequencing, single-cell suspensions were created using tumor dissociated kit and immediately processed for library preparation using 10× Genomics Chromium Chip A/B Single Cell kit and Single Cell GEM, Library & Gel Bead kit (cat. nos. 1000092/100075 and 1000073/1000074). We adhered to the manufacturer protocols as described previously[27]. Single-cell sequencing data were demultiplexed and processed using the 10× cellranger 3.1.0 pipeline to generate the read count matrices for both resistant and sensitive clones. The small fraction of mouse cells were removed by the prediction of gem classification file output by cellranger. Read count matrices were normalized to gene expression measurement with log normalization method using Seurat. Cell states for each condition were assigned using the same clustering resolution and GSEAsig analysis with expert manual curations. The expression matrices from resistant and sensitive clones were integrated using the canonical correlation analysis in Seurat. The differential gene expression analysis was obtained on the same cell state between the resistant and sensitive clones using the FindMarkers functions and thresholded using adjusted $p$ value less than 0.05, absolute average log2 fold change above 0.5 and differential percentage of cells expressing the marker above 0.05. These codes have been uploaded to GitHub repo (https://github.com/qinqian/rms_analysis/tree/master/drug) and Zenodo (DOI:10.5281/zenodo.15767806).

Xenograft SNaPShot and whole-genome sequencing (WGS) were performed after initial expansion, mouse cell depletion, and passaging three times in culture. The SNaPShot technique involves PCR, single-base primer extension, and capillary electrophoresis. Additional details of the methods and analysis used in SNaPShot have been described[94]. For WGS, samples were analyzed by NovaSeq X Plus sequencer (Novogene, 60× coverage) to obtain 150 base pair (bp) paired reads. Samples were processed and analyzed according to the supplemental methods. The average sequencing depth ranged from 60.6× to 69.0× (median depth of 65.7×). The median percentage of the genome covered by at least 20× was 97.1% (range, 94.8%–98.1%). The code for the WGS analysis is available on GitHub at https://github.com/rheinbaylab/Yang_Wang_RMS_resistance.

### Whole genome sequencing

Xenograft whole-genome sequencing was performed using the NovaSeq X Plus sequencer (Novogene) to obtain 150 base pair (bp) paired reads. Read quality statistics were obtained with FastQC (v.0.12.1) (https://www.bioinformatics.babraham.ac.uk/projects/fastqc/). Raw reads were preprocessed using fastp (v.0.20.1) to remove Illumina adapter sequences, and low-quality reads were discarded (Phred quality <20 and less than 51 bp length)[95]. Trimmed reads were aligned to the human UCSC reference genome hg38 using the Burrows−Wheeler Aligner (v.0.7.17)[96]. Potential sources of contamination from other cell lines were assessed using the Calculate Contamination tool from the GATK suite (v.4.2.6.1) (https://broadinstitute.github.io/genomics-in-the-cloud/). To remove read contamination from rare mouse cells, we carried out an additional alignment step to the mouse UCSC genome mm39. Reads aligning unambiguously to the mouse genome were filtered using Xenome (v.1.0.1.r)[97]. BAM file manipulation, including read sorting, was performed using samtools (v.1.11) and the Picard SortSam tool (v.3.1.1) (https://broadinstitute.github.io/picard/)[98]. Duplicated reads were marked using sambamba (v.1.0.1)[99]. Average sequencing depth ranged from 60.6× to 69.0× (median depth of 65.7×). The median percentage of the genome covered by at least 20× was 97.1% (range, 94.8%–98.1%).

### Copy number and structural variant calling

Single-sample (tumor-only) copy number (CN) estimates were generated using CNVkit (v.0.9.10)[100]. Purity and ploidy estimation and paired copy number analysis using the sensitive parental samples as reference was carried out using FACETs[101,102]. We further estimated ploidy and purity and absolute copy changes with the ABSOLUTE method implementation from the PureCN package (v.1.22.2) in R (v.4.1.0)[103,104]. The final CN plots were generated in R using the GenVisR (v.1.30.0) and patchwork (v.1.1.3) packages (https://patchwork.data-imaginist.com/index.html)[105]. SVs, including ≥ 50 bp deletions (DEL), insertions (INS), tandem duplications (DUP), intra- and inter-chromosomal translocations (intraBND and interBND, respectively), and their breakpoints were predicted using manta (v.1.6.0) in comparison to the parental sensitive samples as reference[106]. SVs with a total coverage of <20 reads were filtered out. All CN and SVs with a cell fraction ≥ 30% or a variant allele frequency ≥ 0.3 were considered as potential drivers, respectively.

### Small variant calling

SNVs were called with MuTect1 (v.1.1.7) using the parental sensitive samples as reference[107]. The GRCh38 COSMIC (v.98) database was employed to identify somatic variants, while the dbSNP (v.151) database was utilized for filtering germline variants[108,109]. Variants likely caused by alignment artifacts were filtered from the output VCF file using the FilterAlignmentArtifacts function of GATK. Small INDELs with a maximum length of 49 bp were called by Strelka (v. 2.9.10)[110]. We used ANNOVAR (v.2020-06-07) to annotate SNVs and INDELs[111]. To find SNV and INDEL driver candidates of resistance, we excluded variants with low read depth (<20 reads of sequencing depth), low allele fraction (AF < 0.30), and those present in gnomAD database (genome v.3.1.2) for non-Finnish Europeans with a frequency > 1%[112]. We further selected only non-synonymous variants located in exonic regions and those affecting alternative splicing as potential protein disruptors. The functional effect of these small variants was determined using snpEff (v. 5.2a)[113].

### Statistics and reproducibility

Statistical details are detailed within the methods, results, and figure legends. Bar graph data is shown as mean ± SD (standard deviation). ANOVA, Dunnett test and Student's $T$-tests were used to assess differences where indicated. The Fisher's exact test was also used in some studies to compare values across two samples. A $p$-value of <0.05 was considered statistically significant. Mice were randomly assigned to treatment groups. Numbers of biological and technical replicates are clearly noted throughout the manuscript, most notably within the legends. No data or samples were excluded from the analysis. The Investigators were not blinded to allocation during experiments and outcome assessment, with exception to histology quantifications shown in Figs. 1, 3, and 7. Statistical analysis used Prism 7 (GraphPad).

### Reporting summary

Further information on research design is available in the Nature Portfolio Reporting Summary linked to this article.

## Data availability

The scRNA-seq data generated from mouse tumor samples in this study have been deposited in the Gene Expression Omnibus (GEO), a public functional genomics data repository, under the accession number GSE280546. The Whole Genome Sequencing data generated from the drug-sensitive and resistant samples have been deposited in

the Sequence Read Archive (SRA) database, under the accession number PRJNA1061089. The ChIPseq data presented in Supplementary Fig. 11A have been deposited in the GEO under the accession number GSE274640. All data are included in the Supplementary Information. The raw numbers for charts and graphs are available in the Source Data file. Source data are provided with this paper. Source data are provided with this paper.

## Code availability

All the analysis scripts have been deposited at GitHub and can be accessed using the links: https://github.com/qinqian/rms_analysis/tree/master/drughttps://github.com/rheinbaylab/Yang_Wang_RMS_resistance.Further information on research design is available in the Nature Research Reporting Summary linked to this article.

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

## Acknowledgements

This work was supported by NIH grants R01CA154923 (D.M.L.), R01CA215118 (D.M.L.), R01CA226926 (D.M.L.), R01CA276116 (D.M.L.), R01CA269213 (D.M.L.), U54CA231630 (D.M.L.), DODCA230101 (D.M.L.) and K99CA278696 (Y.Wei). Additional funding included the MGH Research Scholars Program (D.M.L.), Infinite Love for Kids Fighting Cancer Grant (D.M.L.), the Rally Foundation (D.M.L.), The Truth 365 (D.M.L.), CureSearch Acceleration Initiative (D.M.L.), the Summer's Way/Friends of TJ Young Investigator Award (Y. Wei), Tosteson & Fund for Medical Discovery Fellowship from MGH (Y.C., Y. Wei), the Alex's Lemonade Stand Foundation Young Investigator Award (Y.C.), and the Singapore National Research Foundation Fellowship (Y.C.). We thank the MGH Department of Pathology Flow and Image Cytometry Research Core, which has been supported by NIH grants 1S10OD012027-01A1, 1S10OD016372-01, 1S10RR020936-01, and 1S10RR023440-01A1. We also thank Drs. Brendon Manning and Nick Dyson, along with Jeff Hazelwood for discussions on topics pertaining to this manuscript.

## Author contributions

Q.Y., Y. Wang, and C.Y. conceived and designed experiments and wrote the manuscript. Q.Y., Y. Wang, Y. Wei, and S.M. performed experiments and collected data, with assistance from E.A., L.W., P.S., and S.S. L.S. and Q.Q. performed the bioinformatics analyses with supervision by A.J.L. E.R., or L.P. I.O. performed unbiased image quantification and data analysis. D.L. provided supervision and funding and edited the manuscript. All authors provided support in writing the manuscript and gave their approval to the final manuscript version.

## Competing interests

The authors declare no competing interests.

## Additional information

Chuan Yan or David M. Langenau.

