## [Transparent Peer Review file · Nature Communications]

The PIK3CA/AKT pathway drives therapy resistance in rhabdomyosarcoma

Corresponding Author: Dr David Langenau

Version 0:

Reviewer comments:

Reviewer #1

(Remarks to the Author)

Dr. Yang et al from the Langenau lab reports on a finding of PIK3CA pathway upregulation in rhabdomyosarcoma resistance. The findings are important and worthy of clinical translation. Well controlled experiments throughout start with the identification of a promising relapse therapy. With observed resistance, a DNA mutational based hypotheses regarding PARPi/temozolomide resistance in rhabdomyosarcoma is proposed. Without recurrent mutations identified and with reversibility of resistance with serial passage not under PARPi/temo selection, the team investigates resistant cell models in the context of the important single cell work that came out of the same lab and 2 other groups in the past couple years. Again, resistance is not explained by relative enrichment for the identified cell state subtypes. Analyses of the cell state signatures enrich and converge on PIK3CA pathway proteins and the investigators connect this pathway to MDR1 and drug efflux through NRF2. A multitude of orthogonal investigations (overexpression of PIK3C α and siRNA of NRF2 have the predicted effects on efflux and MDR1) across multiple models, including both fusion positive and negative rhabdomyosarcoma models strengthens the evidence and translational potential of this finding. The authors also investigate this mechanism in standard VAC chemotherapy, the front line therapy given to the vast majority of patients at diagnosis with similar findings of Alpelisib inhibition reducing chemotherapy resistance.

Future directions with optimal dosing and schedules are welcome as this promising finding can be translated to sorely needed clinical trials in rhabdomyosarcoma patients. The general sense of several passages to enrich for this pathway upregulation raises several translational questions. My comments are thus mostly in terms of the translational framing of this very promising work.

1. Adjectives for the the clinical response are qualitative rather than quantitative or data supported. Examples include:
 - a. "exceptional responses" intro paragraph 2 which should be framed in terms of response rates or duration of disease control. Typical relapse response rates might be 30% or patients responding and for a duration of a handful of months on average.
 - b. "subset" in the abstract implies the majority of tumors are cured by an intervention which isn't typical in sarcomas. Please reframe. Subset is reused in results as well.
 - c. "completely killed" in results, paragraph 1 might be changed to "complete response"?
 - d. clarify "be effective" perhaps to "have an effect" for OT+alpelisib in regards to VAC R cells (pg 11)
2. The dynamic speed from sensitive to resistant is hinted at throughout the paper and the reversibility is of the utmost importance for translation. The authors do very well to investigate and design experiments to make this overall point clear. In the second to last discussion paragraph, a confusing statement regarding stratifying patients is made. No data is shown that PIK3CA pathway changes prior to therapy are observed or are prognostic in RMS. Might instead the PIK3CA pathway evaluation after some duration of chemotherapy, or some other response based biomarker be more ideal for patient selection? It seems that an introduction of a PIK3Ca inhibitor after some number of cycles of chemotherapy could be considered and many initially low pathway patients would have tumor cells upregulate this pathway under therapeutic selection. In fact the "oscillating" idea in the last paragraph indeed seems to contradict this paragraph 2 suggestion.
3. It isn't clear how the 5 micromolar exposures to Alpelisib compare to what can be achieved in humans, but the murine data makes it clear that tolerable doses have the intended effects. Might you compare the tested doses with human PK? example article <https://doi.org/10.1002/pbc.29897>

4. Alpelisib is FDA approved in children! This should be in the discussion as drug access is typically a barrier in pediatric cancer treatment. While not advocating for this nor clear on how this would be done specifically based on the provided data, it seems warranted to cite and acknowledge that safety data exists for this agent in children: PMID: 37634128 and/or 37624421.

5. In some ways this is a wonderful, and far more robust, rediscovery of MDR1 as chemoresistance in sarcoma. Consider citing prior publications from a generation ago: 22504834 for example.

(Remarks on code availability)

Reviewer #2

(Remarks to the Author)

The resistance of rhabdomyosarcomas to therapies, chemotherapy and radiotherapy, is the main reason for local and systemic progression of the disease. In this manuscript the authors, focusing on the PI3K/AKT pathway, demonstrate how alpelisib, an AKT inhibitor, makes RMS cells sensitive to OT therapy through the inhibition of the expression of ABC transport proteins. The manuscript presents absolutely new and enormously important data. The investigation methods used are absolutely congruent and the data presented correctly. The manuscript, in my opinion, can be published in Nat. Comm.

(Remarks on code availability)

Reviewer #3

(Remarks to the Author)

The article entitled "The PIK3CA/AKT Pathway Drives Therapy Resistance in Rhabdomyosarcoma" provides a comprehensive investigation into mechanisms underlying therapy resistance in rhabdomyosarcoma (RMS), with a particular focus on the PIK3CA/AKT pathway. Both in vitro and in vivo models were established to explore the role of this pathway in mediating resistance to olaparib and temozolomide (OT) combination therapy. Overall, this manuscript is well-structured and thorough. I recommend accepting the paper after the authors successfully address the following comments.

1: The drug-resistant cell lines they used overexpressed three major ABC transporters P-gp, MRP1, and BCRP. Which one plays a major role in drug resistance is not clear. In Fig. 3A, they only showed the Immunofluorescence for MDR1/P-gp, but the Immunofluorescence for BCRP and MRP1 should be shown as well. The Western blotting results in Fig. 3B can not tell if the BCRP and MRP1, like MDR1, are expressed on the cell membranes. Another experiment they can conduct is to use inhibitors of ABC transporters, in their MTT/CCK8 drug sensitivity assay; this is the way to know which pump plays an important role in drug resistance.

2: Several years ago, some researchers reported that inhibiting PI3K may downregulate the expression of P-gp and BCRP. However, this important article was not cited in the paper. The link is: <https://pubmed.ncbi.nlm.nih.gov/35459184/>. In addition, a recent publication about a novel role for CYP1B1 in driving PARPi resistance may be also good to be included in the Introduction section: <https://pubmed.ncbi.nlm.nih.gov/39395328/>

3: There are many abbreviations used throughout the body text and figures. Please add a complete list of abbreviations to ensure clarity for the readers.

4: Some abbreviations appear without being defined on their first use. Please clarify all abbreviations (e.g., "FACS" on Page 6).

5: Page 2, first paragraph: Keep "Fusion-positive (FP) RMS" and "fusion-negative RMS (FN-RMS)" in the same format.

6: Page 2, second paragraph: The term "olaparib PARP inhibitor" should be "PARP inhibitor olaparib".

7: Figure 1: Please clarify why several images were captured after 35 days rather than the full treatment cycle (42 days).

8: Figure 2B: please indicate the meanings of the different colors presented in this figure for clarity.

9: Figure 5A: The quality of these Western blotting images is not satisfied. Please provide some higher-quality images or alternative data.

10: There are some minor grammar errors throughout the text, examples include:

Page 16, "were purchase from Addgene", should be "were purchased".

Page 18, "then take log2 fold change", should be "then taking".

Page 21, "imageJ" should be "ImageJ".

Page 23, "Graphpad" should be "GraphPad".

....

Please carefully check the manuscript for any additional grammatical or typographical errors. It should be better if you can ask an editing service to improve the writing.

(Remarks on code availability)

Reviewer #4

(Remarks to the Author)

Rhabdomyosarcoma (RMS) is the most common soft tissue sarcoma in children and young adults. Despite advances in diagnostic and treatment approaches, 30% of patients eventually relapse due to the development of resistance to the combination therapy of vincristine, actinomycin D, and cyclophosphamide (VAC). This resistance is associated with an abysmal 17% five-year survival rate and occurs irrespective of disease subtype. Gaining a detailed molecular understanding of therapy resistance in RMS will be important for developing effective therapies. In this manuscript, the authors found that a large fraction of olaparib and temozolomide (OT) combination therapy and VAC resistant RMS activate the PIK3CA/AKT pathway to induce NRF2 mediated transcription of multiple multidrug-resistant ABC transporters. They demonstrated that inhibition of PI3K α subunit with alpelisib potently re-sensitized RMS cells to killing by OT through suppressing expression of the ABC transporters and ultimately led to retention of drugs within the cell. However, there are several points/concerns in the manuscript that need to be addressed. Major concern is the lack of rationale for focusing on PIK3CA/AKT pathway.

Major

1. Figure 2G: The authors stated that PIK3CA was upregulated in the RD-D1 and Rh41-R2 models; however, the reviewer couldn't find PIK3CA in Supplementary Table 5 (used to generate Venn diagram shown in Figure 2G). Please provide data to demonstrate the upregulation of PIK3CA in the resistant models. In addition, the rationale for focusing on the PIK3CA/AKT pathway based on scRNA-seq data analysis is not sufficient. Why did authors focus on PIK3CA/AKT pathway regulators, which are detected in only one or two resistant cell models, instead of genes that overlapped across more than three resistant cell models? The reviewer wonders if the single gene that overlaps across all four cell models is more important.

2. Figure 3C: Some assessments of cell viability following drug treatment have not been performed (indicated in gray); thus, the effectiveness of the triple combination therapy cannot be properly evaluated. For example, cell viability in OT-sensitive cells treated with OT + Tariquidar (pan-MDR) is not evaluated; therefore, the effect of this triple combination therapy on OT-sensitive cells is unclear. Additionally, the lack of data for OT-resistant cells treated with Tariquidar monotherapy is not convincing to demonstrate that Tariquidar requires OT to effectively kill resistant tumors. The lack of data makes it difficult to conclude that "pan-ABC transporter inhibitors effectively killed resistant tumors only in the presence of OT and had limited effects on parental, therapy sensitive cells" on page 8, line 15-17.

3. Figure 6: To assess the efficacy of the triple drug combination of alpelisib and OT, the authors used OT-resistant cells (MAST139-R1 and MAST39-R1). Do the authors suggest a sequential treatment strategy (starting triple therapy after acquiring resistance to OT) rather than initiating triple combination therapy from the beginning in OT-naïve cells? Can starting triple combination therapy from the beginning potentially show a durable response in OT-sensitive cells? Please provide the efficacy of initial triple combination therapy in OT-sensitive cells, or discuss the treatment strategy in more detail.

Minor

1. Supplementary Figure 2: The authors described that "two fusion-negative RMS models were effectively killed by OT treatment (381T and RMS559), while the remaining six RMS models developed resistance over time" on page 5, line 1-3. However, it appears that JR-1, RMS176, and Rh3 cells have not yet fully acquired resistance to OT. Please describe it more precisely.

2. Page 5 line 9: "in NSCLC PDX" should be "in non-small cell lung cancer (NSCLC) PDX".

3. Page 6, line 20: "we uncovered a novel, therapy" should be "we uncovered a novel therapy".

4. Supplementary Figure 9: Function analysis using activated PI3K α (H1074R) or myristoylated (myr)-AKT1 is one of the key data in this paper, demonstrating that PIK3CA/AKT pathway is related to ABC transporter expression and function. At least part of Supplementary Figure 9 should be included in the main Figure. Additionally, please consider adding quantitative analyses such as MDR1/GAPDH, MRP1/GAPDH, and BCRP/GAPDH ratios to make the changes clear.

5. Figure 6B: Please provide a scale bar.

(Remarks on code availability)

I did not attempt to run the code, but did review parts of it. The parts I viewed are clear.

Reviewer #5

(Remarks to the Author)

(Remarks on code availability)

Version 1:

Reviewer comments:

Reviewer #1

(Remarks to the Author)

My comments have been fully addressed. Furthermore, changes made and responses to other reviewer comments further strengthens this work.

(Remarks on code availability)

Reviewer #3

(Remarks to the Author)

The authors had addressed the comments accordingly.

(Remarks on code availability)

Reviewer #4

(Remarks to the Author)

The authors have provided additional data and addressed some of the reviewer's questions. I have some additional comments.

1. The rationale for focusing on the PI3K/AKT pathway: In the Response to Referees Letter, the authors state that the PI3K/AKT pathway has been reported to contribute to treatment resistance, and they have incorporated this information into the manuscript. However, before explicitly stating the reason for focusing on the PI3K/AKT pathway, the manuscript first evaluates SNaPshot-based gene mutations and scRNA-seq-based gene expression analysis of genes related to the PI3K/AKT pathway. It implies that the SNaPshot and scRNA-seq analyses were conducted under the assumption that the PI3K/AKT pathway is involved in drug-resistant RMS, which could be confusing for readers. Before describing the results of the SNaPshot (page 6, lines 162-163) or scRNA-seq analysis (page 7, lines 197-200), please clearly explain the rationale for evaluating the PI3K/AKT pathway-related genes.

2. page 6, lines 162-163: No citation is provided to support that PIK3CA, EGFR, HER2, FGFR4, IGF1R, PDGFRA, MET, and ALK genes regulate the PI3K/AKT pathway. Please add appropriate references.

3. page 7, lines 197-200: No citation is provided to support that VEGFA and IGF2 act as mediators of the AKT pathway or that PDGFA, CCL2, CTGF, and IGFBP2 regulate the AKT pathway. Additionally, is there sufficient evidence to conclude that these genes specifically activate the AKT pathway? Please provide references supporting this claim.

4. Figure 3C: Scale bar is truncated, possibly because the figures are cut off at the right.

(Remarks on code availability)

Reviewer #5

(Remarks to the Author)

(Remarks on code availability)

Response to reviewer's comments

Reviewer #1: expert in Rhabdomyosarcoma and AKT

Dr. Yang et al from the Langenau lab reports on a finding of PIK3CA pathway upregulation in rhabdomyosarcoma resistance. The findings are important and worthy of clinical translation. Well controlled experiments throughout start with the identification of a promising relapse therapy. With observed resistance, a DNA mutational based hypotheses regarding PARPi/temozolomide resistance in rhabdomyosarcoma is proposed. Without recurrent mutations identified and with reversibility of resistance with serial passage not under PARPi/temo selection, the team investigates resistant cell models in the context of the important single cell work that came out of the same lab and 2 other groups in the past couple years. Again, resistance is not explained by relative enrichment for the identified cell state subtypes. Analyses of the cell state signatures enrich and converge on PIK3CA pathway proteins and the investigators connect this pathway to MDR1 and drug efflux through NRF2. A multitude of orthogonal investigations (overexpression of PIK3Calpha and siRNA of NRF2 have the predicted effects on efflux and MDR1) across multiple models, including both fusion positive and negative rhabdomyosarcoma models strengthens the evidence and translational potential of this finding. The authors also investigate this mechanism in standard VAC chemotherapy, the front-line therapy given to the vast majority of patients at diagnosis with similar findings of Alpelisib inhibition reducing chemotherapy resistance.

Future directions with optimal dosing and schedules are welcome as this promising finding can be translated to sorely needed clinical trials in rhabdomyosarcoma patients. The general sense of several passages to enrich for this pathway upregulation raises several translational questions. My comments are thus mostly in terms of the translational framing of this very promising work.

Specific comments to address:

1. Adjectives for the clinical response are qualitative rather than quantitative or data supported.

Examples include:

a. "exceptional responses" intro paragraph 2 which should be framed in terms of response rates or duration of disease control. Typical relapse response rates might be 30% or patients responding and for a duration of a handful of months on average.

We have now added quantitative statements describing OT response in preclinical PDX models and clinical trials in the introduction as requested. This revised text is appended below:

"For example, OT treatment reduced tumor growth and significantly prolonged survival of mice xenografted with Ewing's Sarcoma and RMS (Control ~40days vs OT ~100 days). More importantly, OT treatment response for uterine leiomyosarcoma and small cell lung cancer patients was 27% and 41.7%, with median progression free survival of 6.9 and 4.2 months. Indeed, our group has shown that OT was effective in curbing RMS tumor growth in zebrafish and mouse xenografts following three cycles of treatment."

b. "subset" in the abstract implies the majority of tumors are cured by an intervention which isn't typical in sarcomas. Please reframe. Subset is reused in results as well.

We have now deleted and rephrased all instances where "subset" was used to more accurately represent the results.

c. "completely killed" in results, paragraph 1 might be changed to "complete response"?

We have made this change as requested.

d. clarify "be effective" perhaps to "have an effect" for OT+alpelisib in regards to VAC R cells (page 11)

We have modified the text to better frame the results as requested.

2a. The dynamic speed from sensitive to resistant is hinted at throughout the paper and the reversibility is of the utmost importance for translation. The authors do very well to investigate and design experiments to make this overall point clear. In the second to last discussion paragraph, a confusing statement regarding stratifying patients is made. No data is shown that PIK3CA pathway changes prior to therapy are observed or are prognostic in RMS. Might instead the PIK3CA pathway evaluation after some duration of chemotherapy, or some other response-based biomarker be more ideal for patient selection?

We have now rephrased this statement to highlight the potential of using PI3KCA/AKT/NRF2 as markers in VAC resistant patients to stratify them into combination treatment with alpelisib (see modified discussion). Appended below is that revised discussion text:

"Our findings are particularly exciting as this provides a rationale for exploring the deployment of OT + alpelisib in the context of tumors that progress on current chemotherapy regimens and may provide new opportunities to treat refractory disease in the future. Our work also suggests that activation of the PIK3CA/NRF2/ABC transport pathway, which can be assessed using standard IHC on paraffin-embedded sections, might be useful biomarkers for stratifying patients into clinical trials that use combination of alpelisib with either VAC or OT or other salvage therapies commonly used in the refractory/relapse setting. Moreover, PIK3CA pathway activity could be assessed after some duration of VAC chemotherapy and patients with elevated pathway activity treated in combination with alpelisib."

2b. It seems that an introduction of a PIK3Ca inhibitor after some number of cycles of chemotherapy could be considered and many initially low pathway patients would have tumor cells upregulate this pathway under therapeutic selection. In fact, the "oscillating" idea in the last paragraph indeed seems to contradict this paragraph 2 suggestion.

"Oscillating" in this context was used to describe the transcriptionally regulated resistance mechanism observed in our studies where long term passage of cells grown in the absence of OT, revert to a therapy-responsive state (>120 days, n=6 out of 8 models examined, figure 2D-E). This "oscillation" between acquisition and loss of the resistant phenotype is dependent on presence of drug, is transient, and is reversible.

To better illustrate our intended point and to address this important reviewer comment, we have now rephrased the statement to read (see also Discussion):

“Together our results raise the interesting and provocative idea that therapy-resistance in RMS may be commonly regulated by non-genetic mechanisms and cells *can transit* from therapy sensitive to resistant states and vice versa based on chemotherapy pressure, likely accounting for why few genetic mutations have been found in relapse and refractory disease.”

3. It isn't clear how the 5 micromolar exposures to Alpelisib compare to what can be achieved in humans, but the murine data makes it clear that tolerable doses have the intended effects. Might you compare the tested doses with human PK? example article <https://doi.org/10.1002/pbc.29897>

Our choice of 5 μM alpelisib exposure for *in vitro* work was based on preliminary dose finding experiments in RMS cell line models and previously reported *in vitro* studies where doses of alpelisib ranged from 2.5-20 μM , with IC50 approximated at 5 μM [1]. Further, in previously reported studies of Alpelisib using non-RMS sarcoma cell lines, doses up to 50 μM were used and IC50's ranged from 6-18 μM [2, 3].

Our choice for using 50mg/kg of alpelisib *in vivo* was based on published alpelisib xenograft studies and are similar to dosing that can be achieved clinically [3-5]. Alpelisib schedule is based on current clinical trial designs and patient dosing. In patients, most single alpelisib agent clinical trials in pediatric and adult cancer patients use doses ranging from 50mg to 300mg daily [6-8]. The recommended dose of alpelisib for 2–18-year-old pediatric/young adult patients is 50mg/day, but can be gradually titrated to 250mg/day overtime [6, 7, 9]. In adults, a daily, continuous combination of 200-300mg alpelisib and 100-200mg Olaparib is administered to adult ovarian carcinoma patients with tolerable side effects and toxicity [8].

As reported by others in extrapolating dosing from humans to mice, we used the following formula: Mice Dose (mg/kg) \times (weight of mice (kg)/weight of human (kg))^{0.33} = Human Dose (mg/kg) [10]:

Extrapolation based on 2 year old weighing 10kg
 $50\text{mg/kg} \times (0.02/10)^{.33} = 50 \times (.002)^{.33} = 50 \times .129 = 6 \text{ mg/kg (60 mg/day)}$

Extrapolation based on 16-18 year old adolescent weighing 50kg
 $50\text{m/kg} \times (0.02/50)^{.33} = 50 \times (.0004)^{.33} = 50 \times .0756 = 3 \text{ mg/kg (150 mg/day)}$

As noted above, we estimated drug dosing for a 2-year-old at 10kg and 18-year-old at 50kg. Based on our calculations, the mouse doses used are comparable to the FDA approved alpelisib pediatric dosing of 50mg to 250mg daily. Importantly, this dosing of alpelisib and OT was both effective and well-tolerated in curbing tumor growth in the two RMS PDX resistant models analyzed.

We have added clarifying statements to the Results section and Material and Methods section to better describe our dose selection:

Results: "Using similar dosing schedule and concentration of alpelisib as in that trial and used in previous mouse xenograft studies, we engrafted 1×10^6 OT resistant PDX MAST139 or MAST39 subcutaneously into NSG mice and treated mice."

Methods: "Doses administered for Olaparib, temozolomide and alpelisib were selected based on previous published xenograft and clinical trial studies."

4. Alpelisib is FDA approved in children! This should be in the discussion as drug access is typically a barrier in pediatric cancer treatment. While not advocating for this nor clear on how this would be done specifically based on the provided data, it seems warranted to cite and acknowledge that safety data exists for this agent in children: PMID: 37634128 and/or 37624421.

We thank the reviewer for raising this important point. We have added additional text, and the citations mentioned to the revised discussion section.

5. In some ways this is a wonderful, and far more robust, rediscovery of MDR1 as chemoresistance in sarcoma. Consider citing prior publications from a generation ago: 22504834 for example.

We have updated our discussion to cite and to discuss the important findings from this publication (as well as others suggested by additional reviewers) and apologize for their omission in the previously submitted draft.

Reviewer #2: expert in Rhabdomyosarcoma, PIK3CA and NRF2

The resistance of rhabdomyosarcomas to therapies, chemotherapy and radiotherapy, is the main reason for local and systemic progression of the disease. In this manuscript the authors, focusing on the PI3K/AKT pathway, demonstrate how alpelisib, an AKT inhibitor, makes RMS cells sensitive to OT therapy through the inhibition of the expression of ABC transport proteins. The manuscript presents absolutely new and enormously important data. The investigation methods used are absolutely congruent and the data presented correctly. The manuscript, in my opinion, can be published in Nat. Comm.

Thank you for your enthusiastic endorsement of the work. We hope this work will lead to both new insights and possible therapies in the future.

Reviewer #3 (Remarks to the Author): expert in MDR1

The article entitled "The PIK3CA/AKT Pathway Drives Therapy Resistance in Rhabdomyosarcoma" provides a comprehensive investigation into mechanisms underlying therapy resistance in rhabdomyosarcoma (RMS), with a particular focus on the PIK3CA/AKT pathway. Both in vitro and in vivo models were established to explore the role of this pathway in mediating resistance to olaparib and temozolomide (OT) combination therapy. Overall, this manuscript is well-structured and thorough. I recommend accepting the paper after the authors successfully address the following comments.

Specific comments to address:

1) The drug-resistant cell lines they used overexpressed three major ABC transporters P-gp, MRP1, and BCRP. Which one plays a major role in drug resistance is not clear. In Fig. , they only showed the Immunofluorescence for MDR1/P-gp, but the Immunofluorescence for BCRP and MRP1 should be shown as well. The Western blotting results in Fig. 3B cannot tell if the BCRP and MRP1, like MDR1, are expressed on the cell membranes. Another experiment they can conduct is to use inhibitors of ABC transporters, in their MTT/CCK8 drug sensitivity assay; this is the way to know which pump plays an important role in drug resistance.

Thank you for suggesting this important set of experiments. We fully agree with the reviewer that Western blot analysis cannot reveal where the MDR1, MCRP and MRP1 proteins are expressed, but only demonstrates comparative expression levels of these proteins across models. In addition, ABC membrane channels have both active/inactive conformations. When bound and hydrolyzing ATP, channels are active and efflux drugs. By contrast, lack of ATP inactivates these channels. Thus, demonstration of spatial locations of P-gp, MRP1 and BCRP immunofluorescence staining also does not directly indicate if these channels are functioning.

To address this important reviewer comment, we have used the eFluxx-ID green assay and leveraged channel-specific inhibitors, Verapamil (MDR1 inhibitor), MK-571(MRP inhibitor) or Novobiocin (BCRP inhibitor) to assess functional differences in RD and Rh41 sensitive and resistant clones at effluxing substrates. As would be expected, OT resistant clones had lower intracellular levels of the eFluxx-ID® Green dye as compared to sensitive isogenic clones (Supplementary Figure 9A), reflecting that more drug was effluxed from resistant models. Single drug treatment with alpelisib lead to retention of eFluxx-ID® Green dye in cells suggesting that inhibiting PIK3CA inactivates ABC transport function in resistant clones tested (Supplementary Figure 9B,C). Finally, individual treatment with channel-specific inhibitors, Verapamil (MDR1 inhibitor), MK-571(MRP inhibitor) or Novobiocin (BCRP inhibitor) showed partial effects in suppressing drug efflux, in keeping with notion that a wide array of ABC transporters are functionally importantly in regulating resistance. To restate succinctly, there was no dominant ABC transporter that specifically drove resistance in our model.

Supplementary Figure 9. ABC drug transporters are more active in resistant RMS cells when assessed using the eFluxx-ID® Green assay.

A) Flow cytometric analysis of RMS cells were treated with 0.5% DMSO, 20 μ M Verapamil (MDR1 inhibitor), 50 μ M MK-571 (MRP inhibitor) or 50 μ M Novobiocin (BCRP inhibitor) along with the eFluxx-ID® Green dye for 45 minutes. **B)** Representative images and **C)** flow cytometry analysis of Rh41 parental sensitive (S1) and resistant cells (R1, R2) showing the fluorescence eFluxx-ID® signals after treatment with DMSO or alpelisib PIK3CAa inhibitor (AL). Scale bar equals 10 μ m (B).

2) *Several years ago, some researchers reported that inhibiting PI3K may downregulate the expression of P-gp and BCRP. However, this important article was not cited in the paper. The link is: <https://pubmed.ncbi.nlm.nih.gov/35459184/>. In addition, a recent publication about a novel role for CYP1B1 in driving PARPi resistance may be also good to be included in the Introduction section: <https://pubmed.ncbi.nlm.nih.gov/39395328/>*

We thank the reviewer for highlighting these important publications which were inadvertently left uncited in our previous submission. We have now added discussion of these papers to the introduction and discussion to highlight how our work fits in the broader literature and where we have made new contributions.

3) *There are many abbreviations used throughout the body text and figures. Please add a complete list of abbreviations to ensure clarity for the readers.*

We have now added a list of abbreviations on page 25. We also include description of each abbreviation at first description throughout the manuscript.

4) *Some abbreviations appear without being defined on their first use. Please clarify all abbreviations (e.g., “FACS” on Page 6).*

We have clarified the use of abbreviations throughout the revised manuscript, including FACS (Fluorescence-Activated Cell Sorting). We have also ensured that all abbreviations are defined at the time of their first use. Finally, now added a list of abbreviations on page 25.

5) *Page 2, first paragraph: Keep “Fusion-positive (FP) RMS” and “fusion-negative RMS (FN-RMS)” in the same format.*

We have corrected this error accordingly.

6) *Page 2, second paragraph: The term “olaparib PARP inhibitor” should be “PARP inhibitor olaparib”.*

We have corrected this error accordingly.

7) *Figure 1: Please clarify why several images were captured after 35 days rather than the full treatment cycle (42 days).*

IVIS bioluminescence imaging was carried out on 35 days instead of a full treatment cycle of 42 days because the control animal (RD-S1) had to be sacrificed on that day due to tumor burden affecting general health status of animal, in compliance with veterinary recommendations at the time and in alignment with MGH IACUC rules and regulations.

8) *Figure 2B: please indicate the meanings of the different colors presented in this figure for clarity.*

The colors had no meaning. Thus, to stop any confusion in the presentation of these data, we have updated Figure 2B to remove color shading from the Venn. We apologize for the confusion.

9) *Figure 5A: The quality of these Western blotting images is not satisfied. Please provide some higher-quality images or alternative data.*

We have now re-run Western blot experiments in Figure 5A to provide better and higher-quality images as requested. These new blots used the same cell lysates as initially presented and are shown in the revised figure.

10) *There are some minor grammar errors throughout the text, examples include:*

Page 16, “were purchase from Addgene”, should be “were purchased”.

Page 18, “then take log2 fold change”, should be “then taking”.

Page 21, “imageJ” should be “ImageJ”.

Page 23, “Graphpad” should be “GraphPad”.

Please carefully check the manuscript for any additional grammatical or typographical errors. It should be better if you can ask an editing service to improve the writing.

We apologize for these mistakes and have corrected them. We have also comprehensively examined our manuscript for grammar and typo mistakes.

Reviewer #4: expert in WES, scRNA-seq

Rhabdomyosarcoma (RMS) is the most common soft tissue sarcoma in children and young adults. Despite advances in diagnostic and treatment approaches, 30% of patients eventually relapse due to the development of resistance to the combination therapy of vincristine, actinomycin D, and cyclophosphamide (VAC). This resistance is associated with an abysmal 17% five-year survival rate and occurs irrespective of disease subtype. Gaining a detailed molecular understanding of therapy resistance in RMS will be important for developing effective therapies. In this manuscript, the authors found that a large fraction of olaparib and temozolomide (OT) combination therapy and VAC resistant RMS activate the PIK3CA/AKT pathway to induce NRF2 mediated transcription of multiple multidrug-resistant ABC transporters. They demonstrated that inhibition of PI3K α subunit with alpelisib potently re-sensitized RMS cells to killing by OT through suppressing expression of the ABC transporters and ultimately led to retention of drugs within the cell. However, there are several points/concerns in the manuscript that need to be addressed.

1) Major concern is the lack of rationale for focusing on PIK3CA/AKT pathway.

The PIK3CA/AKT pathway is well-known to regulate oncogenesis in a wide array of cancers and in some instances portend worse outcome that can be associated with mutational activation of PIK3CA or inactivation of PTEN. Elevated PIK3CA/AKT pathway activity has also been associated with therapy-resistance in a subset of tumor types. In the context of RMS, we know that IGF1-inhibitor ganitumab prevents the phosphorylation of AKT and synergizes with MEK inhibition to kill FN-RMS [11]. This therapeutic response was observed in models without mutational activation of the PIK3CA/PTEN axis. Moreover, high phosphorylation levels of AKT Ser (473) are associated with poor overall and disease-free survival in a small cohort of RMS patients, further suggesting a link of this pathway in elevating aggression and resistance [12]. These and additional clarifying text have been added to the manuscript to further bolster our focus on assessing the PIK3CA/AKT pathway in the context of RMS resistance. In addition, our wide array of studies supports the role and rationale for studying the PIK3CA/AKT pathway as a dominant resistance mechanism in this disease. These include:

1) 13 of 18 OT-resistant and 6 of 13 VAC-resistant models upregulate the PIK3CA/AKT pathway coincident with acquiring resistance (Figure 3A-B, Supplementary Figure 5 and Figure 6A).

2) Each of these PIK3CA/AKT-pathway active and therapy-resistant models coordinately upregulate the ABC transporters (n=13 out of 13 in OT-resistant models (Figure 3A-B, Supplementary Figure 5); n=6 out of 6 in VAC resistant models (Figure 6A) .

3) Chemical epistasis experiments confirmed that PIK3CA-inhibitor alpelisib suppresses ABC transporter function and synergizes with either OT or VAC to kill resistant models (n=4 of 4 in OT resistant models, Figure 3C; 6 out of 6 in VAC resistant models treated with OT + Alpelisib, Figure 6B, left, and 4 of 6 in VAC resistant models treated with VAC + Alpelisib, respectively, Figure 6B, right, supplementary figure 13).

4) Gain-of-function studies using both activated PI3K α (H1074R) or myristoylated (myr)-AKT1 upregulates *MDR1*, *MRP1* and *BCRP* expression and confers resistance to OT (Figure 4).

5) NRF2 phosphorylation and activation was mediated by PIK3CA/AKT activity (Figure 5 and Supplementary Figure 10). Moreover, loss-of-function studies using siRNA targeting of NRF2 significantly downregulated ABC transporters *MDR1*, *MRP1* and *BCRP* expression and resensitize RD and Rh41 resistant clones to OT treatment (Figure 5A -D, Supplementary Figure 10A-D).

6) Our *in vivo* xenograft studies using Alpelisib and OT significantly reduced tumor burden and extended survival in the two models analyzed (Figure 7).

Together, these data strongly support the rationale for focusing on PIK3CA/AKT pathway as a major driver of resistance in RMS. We have also now provided additional clarifying text and better summary of our scRNA sequencing and chemical screen within the results section which we hope better guides the reader as to how we came to initially focus on this pathway for study,

2) Figure 2G: The authors stated that PIK3CA was upregulated in the RD-D1 and Rh41-R2 models; however, the reviewer couldn't find PIK3CA in Supplementary Table 5 (used to generate Venn diagram shown in Figure 2G). Please provide data to demonstrate the upregulation of PIK3CA in the resistant models. In addition, the rationale for focusing on the PIK3CA/AKT pathway based on scRNA-seq data analysis is not sufficient.

Thank you alerting us to this error. PIK3CA was *not* identified in our initial differential expression analysis of the scRNA sequencing data, which focused on comparing gene expression differences between distinct cell states and across parental vs. resistance models (i.e. identifying genes that were differentially expressed between progenitor cell/cancer stem states for example rather than across all RMS cells). Only after our initial discoveries of PIK3CA being a major driver of RMS resistance did we go back and look at its expression across all RMS cells in our scRNA sequencing data. In address of the reviewer concern in including PIK3CA expression within our scRNA sequencing analysis, we have removed it from our discussion of results and updated the main and supplementary figure to correct this error.

Yet, our studies did identify the upregulation of many important regulators of AKT pathway activity within the scRNA sequencing data as noted in revised Figure 2G and Supplementary Figure 3, 4, which along with well-known roles for this pathway in driving resistance in other tumors and its association elevation in in a small cohort of relapse RMS patients, led us to prioritize inclusion of this pathway in subsequent IHC studies and a mini-drug screen.

Together, the rationale for focusing on PI3KCA/AKT pathway is not only because of the single cell transcriptomic sequencing data, but also based on i) the known importance of the pathway as an oncogenic driver in other cancers, ii) the association of pAKT with aggression and relapse in RMS [12], and iii) supported in part by our scRNA sequencing data that suggested upstream factors known to regulate the AKT pathway (*PDGFA*, *CCL2*, *CTGF* and *IGFBP2*) were upregulated. Yet, to directly, address this important reviewer comment, we have substantially toned down the scRNA sequencing findings and better placed them in context of the known literature and importance of the PIK3CA/AKT pathway in RMS.

Extensive data supporting the upregulation of the PIK3CA/AKT/NRF2/ABC transport pathway as a driver of therapy resistance is detailed throughout the revised manuscript and provided as a summary outlined in reviewer 4, response #1 above.

3) Why did authors focus on PIK3CA/AKT pathway regulators, which are detected in only one or two resistant cell models, instead of genes that overlapped across more than three resistant cell models? The reviewer wonders if the single gene that overlaps across all four cell models is more important.

As noted above, we have toned down our dependence of the scRNA sequencing data on nominating the PIK3CA/AKT pathway for study. Moreover, as correctly noted by the reviewer, the Venn diagram provided in 2G was confusing. In the previous version of the manuscript, we provided both up and down regulated genes in the same Venn-diagram comparison (which is clearly not the right thing to do). In the single case where there was gene overlap, this gene was Cysteine-rich angiogenic inducer 61 (CYR61), a matricellular protein involved in cellular adhesion, migration, proliferation, and survival. While CYR61 has been investigated as a prognostic marker for patient survival in colorectal cancer, not much is known about the role of this protein in RMS. Moreover, CYR61 was upregulated in three models (Rh41, MAST139 and MAST39) while down regulated in one model (RD). Thus, the overlap did not identify a common

up or down regulated gene found consistently across the models (as may have been suggested by our previous rendering of the data).

To better highlight the main findings intended to be shown in Figure 2G, we now provide two Venn diagrams, noting genes that are upregulated across all four models (Figure 2G) and down regulated across all models (Supplemental Figure 3C). We also provide the full list of these gene lists in Supplementary Table 5. Again, the main finding of Figure 2G and Supplementary Figure 3C holds, in that no common up or down regulated genes were identified across the models.

We apologize for the confusion in how we initially presented these combined data.

4) Figure 3C: Some assessments of cell viability following drug treatment have not been performed (indicated in gray); thus, the effectiveness of the triple combination therapy cannot be properly evaluated. For example, cell viability in OT-sensitive cells treated with OT + Tariquidar (pan-MDR) is not evaluated; therefore, the effect of this triple combination therapy on OT-sensitive cells is unclear. Additionally, the lack of data for OT-resistant cells treated with Tariquidar monotherapy is not convincing to demonstrate that Tariquidar requires OT to effectively kill resistant tumors. The lack of data makes it difficult to conclude that “pan-ABC transporter inhibitors effectively killed resistant tumors only in the presence of OT and had limited effects on parental, therapy sensitive cells” on page 8, line 15-17.

We have now performed the additional cell viability experiments as requested (Supplementary Figure 7). As indicated in Figure 3C, single or combination inhibitor treatment for most drugs tested here do not significantly reduce cell death in both sensitive and OT resistant RMS cells. Here, only combination of tariquidar and Alpelisib with OT can robustly potentiate cell killing in OT resistant clones. We append that data below and is also included in the revised manuscript in Supplementary Figure 7.

Supplementary Figure 7. Tariquidar ABC transporter inhibitor re-sensitizes resistant RMS clones to OT treatment. A, D) Tariquidar single drug treatment had minimal effects on tumor cell growth when applied to parental or therapy-resistant RD and Rh41 cells (2.5 μM, indicated by dash line). **B-C, E-F)** Tariquidar (dosed at 2.5 μM) re-sensitized resistant RMS to OT induced killing after treatment for 48 hours. Black asterisks denote significance in comparison to the parental sensitive lines (RD-S and Rh41-S for C and F, respectively) by ANOVA followed by Dunnett post hoc test. Red asterisks denote differences in cell killing by Tariquidar within each model by Student's T-test (C and F). **p<0.01, ***p<0.001 and not significant (ns).

5) Figure 6: To assess the efficacy of the triple drug combination of alpelisib and OT, the authors used OT-resistant cells (MAST139-R1 and MAST39-R1). Do the authors suggest a sequential treatment strategy (starting triple therapy after acquiring resistance to OT) rather than initiating triple combination therapy from the beginning in OT-naïve cells? Can starting triple combination therapy from the beginning potentially show a durable response in OT-sensitive cells? Please provide the efficacy of initial triple combination therapy in OT-sensitive cells, or discuss the treatment strategy in more detail.

We believe the strategy of either sequential treatment or initiating triple drug combination right from the beginning depends on the PI3K/AKT/NRF2 status of the patient receiving treatment. Clinically, most patients that could move onto trials with OT combination therapy would have relapsed disease and prior treatment with VAC. Given we have demonstrated that upregulation of the PI3K/AKT/NRF2 pathway can confer VAC resistance as well, it is likely that some RMS patients would *a priori* have elevated PI3K/AKT/NRF2 pathway activity. In such a scenario, we would advise to initiate triple combination therapy from the beginning. In contrast, patients with

low PI3K/AKT/NRF2 level, we would advise sequential treatment strategy and only adding in alpelisib when resistance would occur.

We have added additional discussion to better describe our proposed strategy of patient stratification based on PI3K/AKT/NRF2 expression (see revised discussion). That text is also appended below:

“Our findings are particularly exciting as this provides a rationale for exploring the deployment of OT + alpelisib in the context of tumors that progress on current chemotherapy regimens and may provide new opportunities to treat refractory disease in the future. Our work also suggests that activation of the PIK3CA/NRF2/ABC transport pathway, which can be assessed using standard IHC on paraffin-embedded sections, might be useful biomarkers for stratifying patients into clinical trials that use combination of alpelisib with either VAC or OT or other salvage therapies commonly used in the refractory/relapse setting. Moreover, PIK3CA pathway activity could be assessed after some duration of VAC chemotherapy and patients with elevated pathway activity treated in combination with alpelisib. Should alpelisib + OT have unexpectedly adverse side effects in future clinical studies, additional clinically available PIK3CA inhibitors have been tested in children including the dual PI3K/mTOR inhibitor samotolisib that was evaluated in the Pediatric ComboMATCH (LY3023414) or copanlisib. ”

6) *Supplementary Figure 2: The authors described that “two fusion-negative RMS models were effectively killed by OT treatment (381T and RMS559), while the remaining six RMS models developed resistance over time” on page 5, line 1-3. However, it appears that JR-1, RMS176, and Rh3 cells have not yet fully acquired resistance to OT. Please describe it more precisely.*

We have amended the results section to reflect the data more accurately.

7) *Page 5 line 9: “in NSCLC PDX” should be “in non-small cell lung cancer (NSCLC) PDX”.*

We have corrected this error.

8) *Page 6, line 20: “we uncovered a novel, therapy” should be “we uncovered a novel therapy”.*

We made the edit as requested.

9) *Supplementary Figure 9: Function analysis using activated PI3K α (H1074R) or myristoylated (myr)-AKT1 is one of the key data in this paper, demonstrating that PIK3CA/AKT pathway is related to ABC transporter expression and function. At least part of Supplementary Figure 9 should be included in the main Figure. Additionally, please consider adding quantitative analyses such as MDR1/GAPDH, MRP1/GAPDH, and BCRP/GAPDH ratios to make the changes clear.*

Thank you for making this important suggestion, which we agree further strengthens the manuscript findings and presentation for the reader. In the revised manuscript, we have moved this supplementary figure into the main text (see revised Figure 4). We have also added quantitative analyses to denote the ratios of MDR1/GAPDH, MRP1/GAPDH, and BCRP/GAPDH, provided in Supplementary Table 7.

10) *Figure 6B: Please provide a scale bar.*

We have added the scale bar as requested (now Figure 7B in the revised manuscript).

Reviewer #4: Remarks on code availability

I did not attempt to run the code, but did review parts of it. The parts I viewed are clear.

Reviewer #5: ECR, co-reviewed with Rev#3

Thank you for your input on this manuscript and suggestions to make it better.

References

1. Piazzzi, M., et al., *Combined Treatment with PI3K Inhibitors BYL-719 and CAL-101 Is a Promising Antiproliferative Strategy in Human Rhabdomyosarcoma Cells*. *Molecules*, 2022. **27**(9).
2. Maeda, M., et al., *In vitro anticancer effects of alpelisib against PIK3CA-mutated canine hemangiosarcoma cell lines*. *Oncol Rep*, 2022. **47**(4).
3. Gobin, B., et al., *BYL719, a new α -specific PI3K inhibitor: single administration and in combination with conventional chemotherapy for the treatment of osteosarcoma*. *Int J Cancer*, 2015. **136**(4): p. 784-96.
4. Kim, K.J., et al., *PI3K-targeting strategy using alpelisib to enhance the antitumor effect of paclitaxel in human gastric cancer*. *Sci Rep*, 2020. **10**(1): p. 12308.
5. O'Brien, N.A., et al., *Targeting activated PI3K/mTOR signaling overcomes acquired resistance to CDK4/6-based therapies in preclinical models of hormone receptor-positive breast cancer*. *Breast Cancer Res*, 2020. **22**(1): p. 89.
6. Remy, A., et al., *Fixed dosing of alpelisib for children with vascular anomalies: Can we do better?* *Br J Clin Pharmacol*, 2025.
7. Singh, S., et al., *FDA Approval Summary: Alpelisib for PIK3CA-Related Overgrowth Spectrum*. *Clin Cancer Res*, 2024. **30**(1): p. 23-28.
8. Konstantinopoulos, P.A., et al., *Olaparib and α -specific PI3K inhibitor alpelisib for patients with epithelial ovarian cancer: a dose-escalation and dose-expansion phase 1b trial*. *Lancet Oncol*, 2019. **20**(4): p. 570-580.
9. Canaud, G., et al., *Alpelisib for treatment of patients with PIK3CA-related overgrowth spectrum (PROS)*. *Genet Med*, 2023. **25**(12): p. 100969.
10. Nair, A.B. and S. Jacob, *A simple practice guide for dose conversion between animals and human*. *J Basic Clin Pharm*, 2016. **7**(2): p. 27-31.
11. Hebron, K.E., et al., *The Combination of Trametinib and Ganitumab is Effective in RAS-Mutated PAX-Fusion Negative Rhabdomyosarcoma Models*. *Clin Cancer Res*, 2023. **29**(2): p. 472-487.
12. Petricoin, E.F., 3rd, et al., *Phosphoprotein pathway mapping: Akt/mammalian target of rapamycin activation is negatively associated with childhood rhabdomyosarcoma survival*. *Cancer Res*, 2007. **67**(7): p. 3431-40.

Reviewer #4:

The authors have provided additional data and addressed some of the reviewer's questions. I have some additional comments.

1. The rationale for focusing on the PI3K/AKT pathway: In the Response to Referees Letter, the authors state that the PI3K/AKT pathway has been reported to contribute to treatment resistance, and they have incorporated this information into the manuscript. However, before explicitly stating the reason for focusing on the PI3K/AKT pathway, the manuscript first evaluates SNaPshot-based gene mutations and scRNA-seq-based gene expression analysis of genes related to the PI3K/AKT pathway. It implies that the SNaPshot and scRNA-seq analyses were conducted under the assumption that the PI3K/AKT pathway is involved in drug-resistant RMS, which could be confusing for readers. Before describing the results of the SNaPshot (page 6, lines 162-163) or scRNA-seq analysis (page 7, lines 197-200), please clearly explain the rationale for evaluating the PI3K/AKT pathway-related genes.

We have added additional clarifying text to further explain the rationale for evaluating the PI3K/AKT pathway-related genes in advance of performing the SnaPshot and scRNAseq studies and included additional citations.

2. page 6, lines 162-163: No citation is provided to support that PIK3CA, EGFR, HER2, FGFR4, IGF1R, PDGFRA, MET, and ALK genes regulate the PI3K/AKT pathway. Please add appropriate references.

We have revised this portion of the manuscript and added the appropriate references as requested.

3. page 7, lines 197-200: No citation is provided to support that VEGFA and IGF2 act as mediators of the AKT pathway or that PDGFA, CCL2, CTGF, and IGFBP2 regulate the AKT pathway. Additionally, is there sufficient evidence to conclude that these genes specifically activate the AKT pathway? Please provide references supporting this claim.

We appreciate the reviewer's comment and have now provided supporting references to justify the connection between these genes and the PI3K/AKT pathway. Multiple studies have demonstrated that VEGFA, IGF2, PDGFA, CCL2, CTGF, and IGFBP2 either activate or regulate the PI3K/AKT signaling cascade:

- **VEGFA:** VEGFA primarily signals through VEGFR-2, which upon activation recruits PI3K and leads to downstream AKT phosphorylation. (PMID: 22647379)
- **IGF2:** IGF2 has been shown to directly activate AKT phosphorylation via the PI3K pathway, as demonstrated by protein analysis in neural precursors. (PMID: 18675921)

- **PDGFA**: PDGFA binds to PDGFR and activates multiple downstream pathways, prominently including PI3K/AKT/mTOR signaling. GO enrichment analysis also confirms its role in activating PI3K/AKT. (PMID: 35105853)
- **CCL2**: Knockdown of CCL2 expression was shown to suppress PI3K/AKT signaling activity, indicating its regulatory role in this pathway. (PMID: 36795287)
- **CTGF**: CTGF promotes AKT phosphorylation and nuclear translocation through activation of the PI3K/AKT pathway, while its inhibition leads to reduced PI3K/AKT signaling. (PMID: 33522639)
- **IGFBP2**: IGFBP2 modulates PI3K/AKT signaling indirectly through its interaction with PTEN, a well-known inhibitor of this pathway. Increased IGFBP2 levels are associated with reduced PTEN activity and enhanced PI3K/AKT signaling. (PMID: 31925333)

While we acknowledge that the involvement of these genes may vary across cell types and biological contexts, the cited literature provides substantial evidence supporting their regulatory roles—either directly or indirectly—on the PI3K/AKT pathway. We have now included these references in the revised manuscript to clarify and support this point.

4. Figure 3C: Scale bar is truncated, possibly because the figures are cut off at the right.

The scale bar has been fixed.